# Performance evaluation of micro lens arrays: Improvement of light intensity and efficiency of white organic light emitting diodes

Apurba Adhikary[1,2], Joy Bhuiya[1], Saydul Akbar Murad[1,3], Md. Bipul Hossain[1], K. M. Aslam Uddin[1], MD Estihad Faysal[4], Abidur Rahaman[1], Anupam Kumar Bairagi[5]*

1 Information and Communication Engineering, Noakhali Science and Technology University, Noakhali, Chittagong, Bangladesh, 2 Department of Computer Science and Engineering, Kyung Hee University, Yongin-si, Republic of Korea, 3 Faculty of Computing, Universiti Malaysia Pahang, Pekan, Pahang, Malaysia, 4 Computer Science and Technology, Henan Polytechnic University (HPU), Jiaozou, Henan, China, 5 Computer Science and Engineering Discipline, Khulna University, Khulna, Bangladesh

* anupam@ku.ac.bd

**Data Availability Statement:** All relevant data are within the paper.

## Abstract

This paper proposes a unique method to improve light intensity and efficiency of white organic light emitting diodes (OLEDs) by engraving micro lens arrays (MLAs) on the outer face of the substrate layer. The addition of MLAs on the substrate layer improves the light intensity and external quantum efficiency (EQE) of the OLEDs. The basic OLED model achieved an EQE of 14.45% for the effective refractive index (ERI) of 1.86. The spherical and elliptical (planoconvex and planoconcave) MLAs were incorporated on the outer face of the substrate layer to increase the EQE of the OLEDs. The maximum EQE of 17.30% was obtained for Convex-1 (elliptical planoconvex) MLA engraved OLED where the ERI was 1.70. In addition, Convex-1 MLA engraved OLED showed an improvement of 3.8 times on the peak electroluminescence (EL) light intensity compared to basic OLED. Therefore, Convex-1 MLA incorporated OLED can be considered as a potential white OLED because of its excellent light distribution and intensity profile.

## 1 Introduction

Organic electronics have become a topic of technological interest since the 1950s starting with the studies of organic crystals as potential crystalline semiconductors [1]. A close perception of organic materials started in the 1950s, had changed it as a potential candidate for semiconducting and even conducting material which was primarily considered as a pure electrical insulator. After the detection of electroluminescence behavior in organic material, it was also detected in inorganic materials [2]. Electroluminescence is the outcome of radiative recombination between an optical phenomenon and electrical phenomenon where materials radiate lights in response to the passage of an electric current and the intensity of the radiated lights in this event is called electroluminescence intensity. The electroluminescence intensity has become an important parameter in today's lighting technology where nearly 50% of the total

**Funding:** The author(s) received no specific funding for this work.

**Competing interests:** The authors have declared that no competing interests exist.

energy is being used to produce light. In addition, the demand for lighting technology has been increasing day by day which requires more energy. Thus, light extraction efficiency, out-coupling efficiency, or intensity improvement of light needs to be improved for better performance of OLEDs. An OLED consists of a reflective cathode, organic layers, a transparent anode, and a substrate (glass). Injected holes and electrons from anode and cathode, correspondingly, traverse over organic layers i.e. electron transport molecular layer (ETML) and hole transport molecular layer (HTML), respectively, and create an attached pair in the emissive layer. The attached pair produces a photon by decaying radioactively and is extracted from the surface of the substrate [3]. As a solid-state, full-color capable, electroluminescent, and flexible device, OLEDs has stepped up the display technology in the commercial market. OLEDs have created a new and sensational display technology by taking the benefits of the self-emitting property, low power consumption, faster-switching speed, and wide viewing angle. OLEDs produce electrically driven lights in non-crystalline organic materials. Due to the potential uses of OLEDs in displays and solid-state illumination, lots of investigation has been observed for expanding the efficiency and operation of the OLEDs. Organic electroluminescence devices require high voltages and produce low efficient lights which are responsible for the slow development of organic electroluminescence devices. The conventional OLEDs experiences certain problems including lower light extraction or out-coupling efficiency and minor viewing angle [4]. The extreme volume of light is missing inside the OLED device owing to various mechanisms which are responsible for the trapping of light and lower light extraction efficiency of OLED as shown in Fig 1 [5]. In addition, about 30% and 20% of light can be trapped due to the surface plasmons means in organic layers and anode/organic wave-guided modes in the anode layer, respectively [6]. Another major trapping of light (about 30%) is detected at the air/substrate interface owing to the TIR (total internal reflection) which causes the confinement of generated lights inside the OLED organic layers and in the interface of the substrate and transparent conducting oxide. For all the reasons, a small fraction of light is emitted in the air from the OLED device. Consequently, it is mandatory to solve the small out-coupling efficiency of light affected by TIR. Numerous research and studies are carried out to resolve the effect of TIR and improve the light extraction efficacy and light intensity of OLEDs. The investigation includes introducing a texturing layer on the substrate layer [6, 7], micro/nano shapes on the top of the substrate layer [8, 9], mesh pattern [10, 11], photonic crystals configuration [12–14] and adding micro/nano-lens or pillars in the substrate layer [4]. The first movement of electroluminescence from natural devices which existed in ac form was started by a set of investigators in 1953 [15]. After 10 years, Pope and Coworkers surveyed the report on solitary crystal anthracene OLEDs [2]. However, Ching W. Tang et al. confirmed the first competent OLED in 1987 which had the EQE of 1% [16]. The luminous efficiency of the first competent OLED was 1.5 lm/W, both EQE and luminous efficiency were very poor. Kido et al. proposed the first white OLED which showed white light emission in the visible region having 3400 $Cd/m^2$ luminance maximum using a double emitting layer structure [17]. K. j. Ko et al. proposed a technique using BaTiO3 embedded substrate which produced OLED light of 1.6 times higher EQE and luminance efficiency than basic design [18]. S. J. Park et al. proposed an OLED by using a Si3N4-based photosensitive scattering layer which enhanced the EQE on 10,000 $Cd/m^2$ paralleled to basic OLED structure [8]. A. Kumar et al. suggested a method of producing OLED by applying nanostructured ITO sandwiched between anode and substrate which increased the beam extraction efficacy [3]. V. Mann et al. recommended a system to incorporate an insulator nanoparticle layer on the substrate of the OLEDs to improve the beam extraction efficacy and increased the OLED efficacy by a factor of 1.7 times associated to conventional OLED [5, 19]. J. G. Kim et al. offered rectangular and hexagonal structures-based OLED whose EQE was expanded considerably [20]. S. Hassan et al. developed an OLED

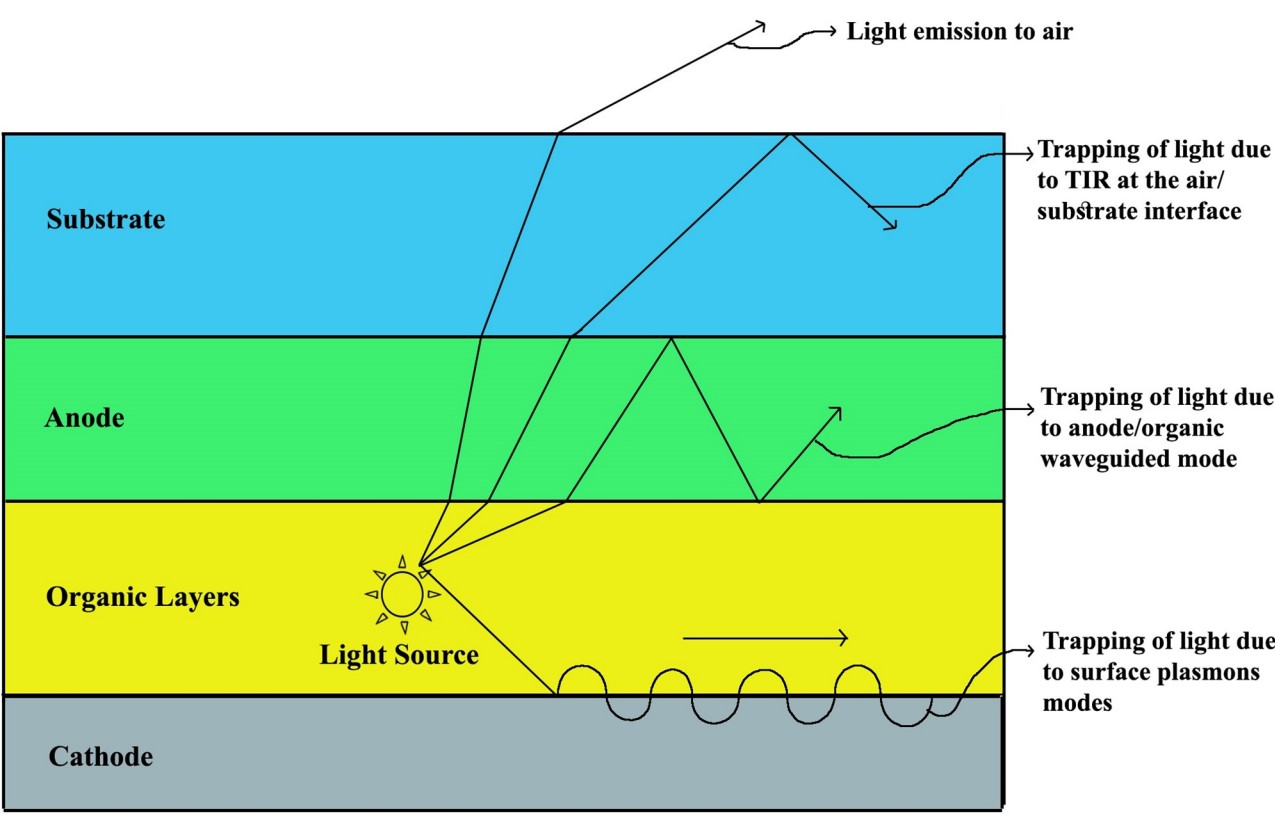

**Fig 1. OLED model illustrating trapping of light due to various modes [5].**

together with enhanced beam extraction efficacy by means of double periodicity along with the graded super-crystals pattern in the cathode [21]. B. K. Kong et al. used Bathophenanthroline (Bphen) particles as a scattering layer to increase the electroluminescence (EL) intensity of YOLEDs (Yarn-based OLEDs) and increased the EL intensity about 1.3 times compared to without Bphen particles incorporated YOLEDs [22]. C-L Huang et al. used photochemical reduction method for the modulation of the shape of silver nanoparticles and improved the efficiency of the polymer light-emitting diodes [23].

Recent OLED based works gives importance on increasing the photosensitive performance of OLEDs. The most significant approaches of performance improvement of OLEDs are enhancing the light extraction efficiency and light intensity. EQE and IQE i.e. internal quantum efficiency of the OLEDs are needed to be improved for the overall improvement of the light extraction efficiency and light intensity. Proper choice of materials leads the higher internal quantum efficiency of OLEDs. In order to select proper raw materials of OLEDs in every single layer, refractive index and lattice constant matching in the adjacent layers are important. On the contrary, EQE can be improved by using micro structuring in the substrate of the OLED. However, it is challenging to improve the EQE and IQE by means of the conventional technique. So far, OLED-based research works for the improvement of EQE and light intensity using micro/nano structuring are limited. On the other hand, different micro/nano structuring techniques were used for the design of different photonic devices. Thus, we can predict that OLEDs might have a great future in structuring techniques for the improvement of EQE and light intensity. Therefore, additional studies are needed for OLEDs to optimize the microstructure technique.

In this work, we incorporated an array of planoconvex and planoconcave lenses at the substrate layer and achieved high EQE and electroluminescence (EL) light intensity. We selected proper materials for each layer of the OLEDs considering refractive index and lattice constant matching among the adjacent layers. The produced OLED lights reflected backwards onto the OLED device from the interface of air/substrate because of TIR. This kind of behavior of OLED lights occurred since substrate has higher refractive index compared to air. Consequently, we engraved micro lens arrays on the substrate layer to reduce the TIR. The incorporation of MLAs on the outer surface accelerates the emitted lights from the surface to intrude with each other and create constructive patterns and destructive patterns in numerous locations. These patterns and distance between them reveal the smoothness and light spreading of the generated OLEDs. Therefore, various shapes and sizes of micro lenses have different impacts on the constructive and destructive patterns. In addition, for the amalgamation of MLAs on the OLEDs, maximum incident angles of light became smaller compared to the critical angle because of curvature shape of the lenses on the substrate layer and led the generated light to be refracted from the outer surface. As a result, EQE and intensity of OLED lights can be increased. We used FDTD (finite difference time domain) method for designing OLEDs. The simulation results confirmed that our proposed technique can produce an EQE of 17.30% for the Convex-1 elliptical planoconvex lens engraved OLED. In addition, EL light intensity is significantly increased for Convex-1 MLA engraved OLED compared to basic OLED. Outstanding light distribution and intensity in the visible range are observed for Convex-1 MLA engraved OLED which guarantees the production of white OLED. We believe that our proposed light intensity and efficiency improvement method will be applicable for the production of efficient OLED.

## 2 Basic concepts of organic light emitting diode

A massive evolution has been achieved in the field of optoelectronic devices, organic FETs and OLEDs for the purpose of switching activities. An OLED has at least one layer having semiconducting organic molecules which is inserted between two-fold electrodes. Anode is selected as a transparent material and cathode is selected as a metal (transparent or non-transparent) for the design of OLEDs. Organic layers such as ETML (emissive layer), HTML (conductive layer), and substrate together with cathode and anode are the essential layers of an OLED [7]. The substrate needs to be transparent and made of glass. All other layers are deposited on the glass substrate whereas the anode layer is responsible for taking away electrons when the required voltage is applied to flow current into the structure. Organic layers which are made of small organic molecules act as conducting and emissive layers for an OLED. Organic hole transport molecules acting as a conducting layer is liable for removing electrons out from the transparent anode of OLED. On the contrary, organic electron transport molecules act as an emissive layer, responsible for carrying electrons from the cathode layer and producing light. Hence, the cathode layer is responsible for injecting electrons into the organic emissive layer as soon as forward biasing voltage is applied. IQE and EQE are important parameters that are required to achieve desired results and minimize the waste of valuable resources i.e. time, energy, and physical materials. IQE depends on the architecture of the device and the charge recombination process. When a forward electrical current/ biasing voltage is employed between cathode and anode, holes are brought into the conductive layer from the anode, and electrons are brought into the emissive layer from the cathode. Holes reach the interface of the emissive layer (ETML) and increase the band frame divergence in the electron transport layer by making electron-hole pairs. The electron-hole pairs produce two types of recombination, for instance, radiative recombination and non-radiative recombination. Radiative recombination produces light and non-radiative recombination generates heat in the form of loss.

The link between the two organic layers (conductive and emissive layer) delivers an effective spot for the radiative recombination of electron-hole pairs and ensuing electroluminescence. Thus, radiative recombined electron-hole pairs determine the value of IQE which can be stated as the fraction of radiative recombination ratio to the entire recombination ratio. Eq 1 can be used to define IQE (iqe) if the radiative and non-radiative recombination ratios are represented separately by Rr and Rnr.

$$\eta_{iqe} = \frac{R_r}{R_r + R_{nr}} \tag{1}$$

The EQE, or the percentage between the total number of photons discharged from the OLED and the total number of photons generated inside the OLED, is the most essential statistic for evaluating OLED performance. In a different logic, we can say that EQE of OLED is the proportion of generated light that is capable to go outside the OLED device. The EQE ($\eta_{eqe}$) can also be written as the product of ($\eta_{iqe}$) and the out-coupling efficiency ($\eta_{out}$), (See Eq 2 [24])

$$\eta_{eqe} = \eta_{iqe} * \eta_{out} \tag{2}$$

where ($\eta_{out}$) depends on the direction and propagation of the external modes i.e. light that can be able to escape from the surface of the substrate. Achieving efficiency close to unity is an important task to optimize the parameters. The light extraction efficiency or EQE of OLEDs can also be estimated not only for a smooth substrate surface but also for an optimized cavity into the substrate surface. The light extraction efficiency, hence, EQE of OLEDs for a smooth substrate and optimized cavity substrate surface can be expressed as in Eqs 3 and 4 respectively [25].

$$\eta_{ext} = 0.5 * n^{-2} * 100\% \tag{3}$$

$$\eta_{ext} = 0.75 * n^{-2} * 100\% \tag{4}$$

Here n symbolizes the effective refractive index (ERI) of the OLEDs.

## 3 Materials and methods

In this work, we designed a basic structure of OLED without any light scattering structure. Afterward, we designed OLEDs with higher intensity by incorporating different types of micro lenses on the outer face of the substrate layer. The physical parameters of the micro lenses were varied to increase the light out-coupling efficacy, which in order increased the overall light intensity of the OLEDs. The OLEDs were designed, and their photosensitive properties were simulated in the 3D version of OptiFDTD (Version 16.0) simulator. The OptiFDTD software system is based on the FDTD (finite-difference time-domain) method. We considered three observation areas (Near field—0 μm, far field 1—6 μm, far field 2—10 μm) from the end face of the substrate to analyze the simulation results where beam profile, the light intensity with other optical properties were investigated. In our research work, we have replaced near field, far field 1, and far-field 2 by OP_1, OP_2, and OP_3 respectively where OP describes the Observation Point.

### 3.1 Materials properties

Our proposed OLED consists of the following layers: anode, cathode, electron-transport molecular (emissive) layer, hole-transport molecular (conductive) layer, and substrate. The

properties of the materials used in different layers of OLEDs during our simulation are given below:

**Anode**: Because of its widespread acceptance as an anode material, we utilized ITO (indium tin oxide) in our design [26]. Indium tin oxide is highly transparent to visible light, has easy patterning ability, and good connection to the glass substrates.

**Cathode**: In OLEDs, stable metals like aluminum (Al) and silver (Ag) are popular as cathodes [27]. In our design, we considered aluminum (Al) as a cathode owing to its higher electrical conductivity.

**Hole-transport molecular layer (HTML)**: Materials along with electron-contributing properties operate as hole-transporting materials. We considered $\alpha$-NPD (N,N'-di(1-naphthyl)-N,N diphenyl-(1,1'-biphenyl)-4,4'-diamine) as HTML [7].

**Electron-transport molecular layer (ETML)**: Materials having electron-accepting properties are used as electron-transporting materials in OLEDs. In our design, we used Tris (8-hydroxyquinoline) aluminum i.e. ($Alq_3$) as electron-transporting material because it is thermally and morphologically stable [28]. As an emissive layer that transfers electrons from the cathode layer and generates light, ETML is important.

**Substrate**: In our design, we used silicon-di-oxide ($SiO_2$) as substrate. Various micro lenses have been incorporated in the substrate surface of the proposed OLEDs to improve the out-coupling efficacy i.e. external quantum efficiency. In addition, we used the same material of ($SiO_2$) as used in the substrate for micro lenses i.e. for micro-structuring of OLEDs.

We considered the physical properties of several materials for different layers of the OLEDs and selected the materials by refractive index and lattice constant matching among adjacent layers. As stability and efficiency of OLED device is important, we achieved the optimized stability and efficiency by considering α-NPD and Alq3 as the donor i.e., electron-contributing and acceptor i.e., electron-accepting material in the emissive layer of OLEDs, correspondingly. Thus, we believe that our designed OLED will be stable and provides maximum performance which will lead to a great contribution in the field of OLEDs. We also consider different size and shape of the micro lenses engraved structures which are incorporated in the substrate layer. Since our design and simulations were conducted in 3D version of OptiFDTD simulator, we considered fixed length of materials for each of the layers of the OLEDs and considered different thickness for the materials. The physical parameters of the OLED layers are summarized Table 1.

## 3.2 Our proposed OLEDs' design parameters

In this work, we began our research considering a basic OLED formation i.e. no micro lens array (MLA) was incorporated in the substrate as illustrated in Fig 2(a). We considered two parts i.e. primary part and the second part in our OLED design. The primary part includes cathode, ETML, HTML, anode, and the second part includes a substrate that is responsible for supporting the OLED design. After designing a primary OLED structure, we incorporated various sizes and shapes of lenses such as spherical and elliptical lenses (planoconvex and plano-concave) in the outer face of the substrate layer, as described in Fig 2(b). Due to the smooth substrate of the basic OLED structure (see Fig 2(a)), trapping of light is occurred inside the substrate for the reason of total internal reflection (TIR) at the air/substrate interface, as illustrated in Fig 2(c). For the reduction of TIR at the air/substrate boundary, an array of micro lens (see Fig 2(b)) is added at the outer face of the substrate, which in turn improves the EQE

**Table 1. Physical properties of several layers of OLEDs [29].**

| Material | Layer name | Length ($\mu m$) | Thickness ($\mu m$) | Refractive Index | Lattice Constant (Å) |
|---|---|---|---|---|---|
| Al | Cathode | 10 | 0.15 | 1.59 | 4.046<br>a = 13.5190 |
| ($Alq_3$) | Electron-transport molecular layer (ETML) | 10 | 0.18 | 1.75 | b = 15.8550<br>c = 18.7110 |
| $\alpha$—NPD | Hole-transport molecular layer (HTML) | 10 | 0.6 | 1.9 | a = 7.9125<br>c = 4.78382 |
| ITO | Anode | 10 | 0.2 | 1.8 | 10.1247<br>a = 4.914 |
| ($SiO_2$) | Substrate | 10 | 0.9 | 1.45 | c = 5.405 |

along with light extraction efficiency i.e. overall efficiency of the micro lens incorporated OLEDs, as shown in Fig 2(d). Fig 2(d) indicates the production of more lights from the OLEDs which would generally be trapped in the substrate layer (see Fig 2(c)) due to the addition of micro lens array on the substrate layer by means of the reduction of the TIR at the air/substrate boundary.

We considered spherical and elliptical lenses for the simulation. In the case of purely spherical lenses, both radius and height had the same value. Consequently, the height-to-radius ratio

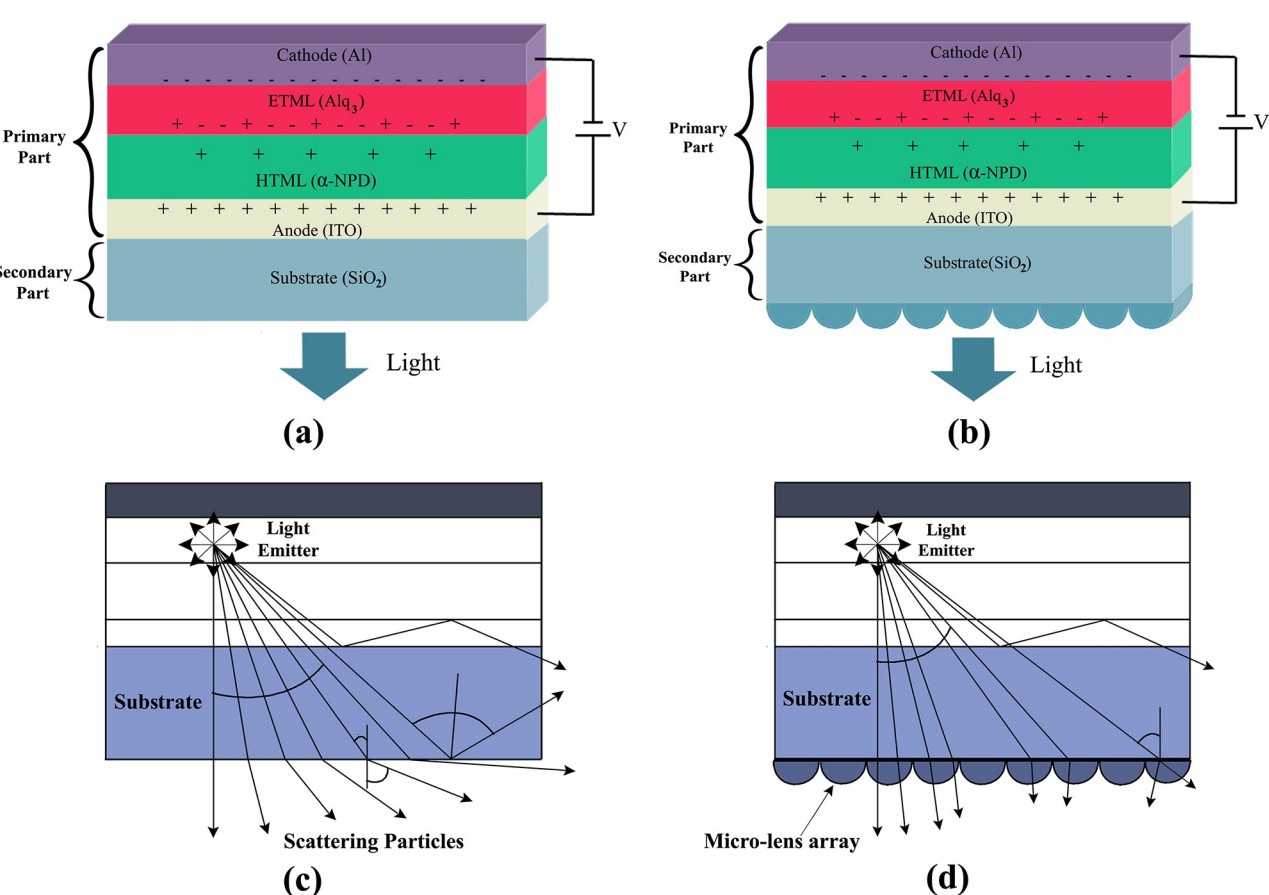

**Fig 2.** (a) Schematic diagram of primary OLED model; (b) Schematic illustration of micro lens array engraved OLEDs; (c) Light trapping in the substrate layer of basic OLED due to total internal reflection [30] and Improvement of light efficiency and intensity of OLEDs using MLA [30].

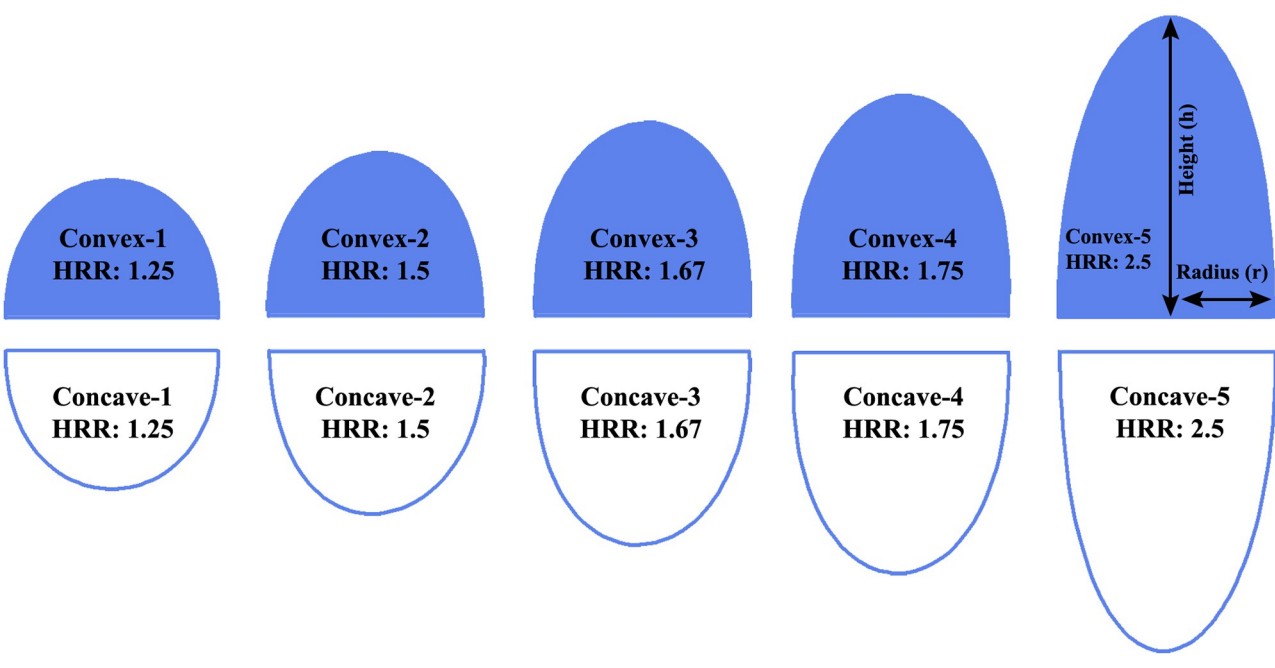

**Fig 3. Different types of elliptical micro lenses engraved on the top of the substrate of the OLEDs.**

(HRR) of all the spherical lenses(SL) was always 1. We used both planoconvex spherical lenses (Spherical-1 to Spherical-4) and planoconcave spherical lenses (Spherical-5 to Spherical-8) in our simulation. On the contrary, we considered five different types of planoconvex elliptical lenses (Convex-1 to Convex-5) and planoconcave elliptical lenses (Concave-1 to Concave-5) during the simulation of the OLEDs, as illustrated in Fig 3. Furthermore, the minor axis radius and main axis radius were determined as the lenses' radius (r) and height (h), respectively.

The values of 1.5, 1.25, 1.75, 1.67, and 2.5 were the HRR of the Convex-2 and Concave-2, Convex-1 and Concave-1, Convex-4 and Concave-4 and Convex-5, Convex-3 and Concave-3, and Concave-5 lenses, correspondingly. The micro lenses have the radius of curvature (R) which can be obtained by Eq 5.

$$R = \frac{r^2 + h^2}{2h} \tag{5}$$

where r and h represent the radius and height of the micro lens. Eq 6 of the lens maker's equation can be used to get the focal length (f) of lenses built of ($SiO_2$) if n is the refractive index.

$$f = \frac{R^2 + h^2}{2h(n-1)} - h(n-1) \tag{6}$$

### 3.3 Design and models of various OLEDs

We started our research by designing a basic OLED structure having no micro lens array on the outer surface of the substrates. Fig 4(a) and 4(d) illustrate the 2D (two dimensional) layout presentation and 2D refractive index profile (RIP) distribution of the white OLED primary model, respectively. We added MLA on the outer surface of the substrate layer to enhance light extraction and light intensity efficiency. We considered various sizes and shapes (planoconcave and planoconvex) of spherical and elliptical MLA for the design and simulation of OLEDs. We considered the same physical parameters of the spherical and elliptical micro

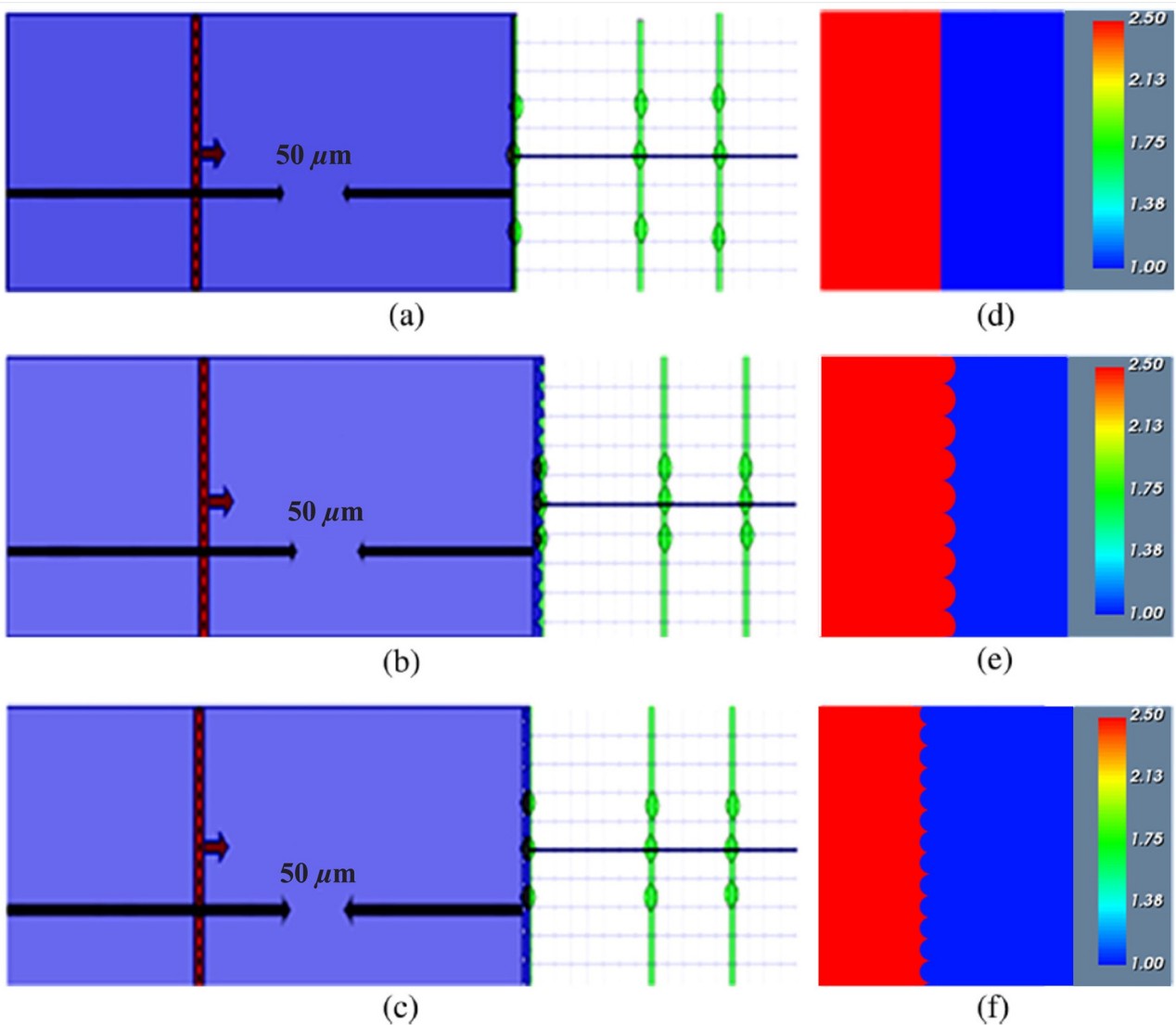

**Fig 4.** (a)-(c) 2D layout presentation and (d)-(f) RIP of different kinds of OLEDs designs: (a), (d) Basic OLED model; (b), (e) Elliptical planoconvex MLA incorporated OLED; (c), (f) Spherical planoconcave MLA incorporated OLED.

lenses for OLEDs as studied for our other work of LEDs [31]. Fig 4(b) and 4(e) demonstrate the 2D layout representation and 2D RIP of the designed OLED with planoconvex elliptical MLA incorporated substrate. The two-dimensional layout presentation and RIP of the planoconcave spherical MLA engraved OLED is illustrated in Fig 4(c) and 4(f), separately. In addition, Fig 4(d)–4(f) represent the red color for the RIP of the substrate and the blue color for the RIP of air.

Planoconcave and Planoconvex lenses had their radius increased from 0.4 microns to 0.7 microns, resulting in an HRR of 1. In the case of elliptical lenses i.e. planoconvex lens and planoconcave lens, major axis, and minor axis are considered as lens height and lens radius, respectively. In the beginning, we varied the lens radius from 0.2 $\mu$m to 0.4 $\mu$m, keeping the lens height fixed at 0.5 $\mu$m. Afterward, we varied the lens height from 0.5 $\mu$m to 0.7 $\mu$m and fixed the lens radius to 0.4 $\mu$m. As a result, elliptical MLA has molded the HRR value ranging

**Table 2. Physical properties of several layers of OLEDs [29].**

| Structure Type/Lens Shape | Lens Type & Lens Height (h) and Lens Radius (r) (μm) | Height-to- Radius Ratio (HRR) | Radius of Curvature (R) (μm) | Focal Length (f) (μm) | Lattice Constant (Å) |
|---|---|---|---|---|---|
| Spherical Planoconvex Lens | Spherical—1 | h: 0.4; r: 0.4 | 1 | 0.4 | 0.71 |
| | Spherical—2 | h: 0.5; r: 0.5 | 1 | 0.5 | 0.89 |
| | Spherical—3 | h: 0.6; r: 0.6 | 1 | 0.6 | 1.06 |
| | Spherical—4 | h: 0.7; r: 0.7 | 1 | 0.7 | 1.24 |
| Spherical Planoconcave Lens | Spherical—5 | h: 0.4; r: 0.4 | 1 | -0.4 | -0.71 |
| | Spherical—6 | h: 0.5; r: 0.5 | 1 | -0.5 | -0.89 |
| | Spherical—7 | h: 0.6; r: 0.6 | 1 | -0.6 | -1.06 |
| | Spherical—8 | h: 0.7; r: 0.7 | 1 | -0.7 | -1.24 |
| Spherical Planoconcave Lens | Spherical—5 | h: 0.4; r: 0.4 | 1 | -0.4 | -0.71 |
| | Spherical—6 | h: 0.5; r: 0.5 | 1 | -0.5 | -0.89 |
| | Spherical—7 | h: 0.6; r: 0.6 | 1 | -0.6 | -1.06 |
| | Spherical—8 | h: 0.7; r: 0.7 | 1 | -0.7 | -1.24 |
| Elliptical Planoconvex Lens | Convex—1 | h: 0.5; r: 0.4 | 1.25 | 0.41 | 0.7 |
| | Convex—2 | h: 0.6; r: 0.4 | 1.5 | 0.43 | 0.74 |
| | Convex—3 | h: 0.5; r: 0.3 | 1.67 | 0.34 | 0.59 |
| | Convex—4 | h: 0.7; r: 0.4 | 1.75 | 0.46 | 0.8 |
| | Convex—5 | h: 0.5; r: 0.2 | 2.5 | 0.29 | 0.52 |
| Elliptical Planoconcave Lens | Convex—1 | h: 0.5; r: 0.4 | 1.25 | -0.41 | -0.7 |
| | Convex—2 | h: 0.6; r: 0.4 | 1.5 | -0.43 | -0.74 |
| | Convex—3 | h: 0.5; r: 0.3 | 1.67 | -0.34 | -0.59 |
| | Convex—4 | h: 0.7; r: 0.4 | 1.75 | -0.46 | -0.8 |
| | Convex—5 | h: 0.5; r: 0.2 | 2.5 | -0.29 | -0.52 |

from 1.25 to 2.5. We summarized the physical parameters of the spherical and elliptical micro lenses added in the substrate of the OLEDs for structuring in Table 2.

## 3.4 Simulation and analysis

A large number of white OLEDs were designed and simulated in the 3D version of the OptiFDTD 64 bit [Version: 16.0] simulator. We investigated different photosensitive features for instance electroluminescence (EL) spectrum (intensity), 2D image map, and 3D distribution of the image of the discrete Fourier transform (DFT) in the y-direction (Ey) of the electric field component. To compute light extraction efficiency, the effective refractive index from the simulation of basic OLEDs and various MLA etched OLEDs is used to determine the external quantum efficiency. We investigated the electroluminescence (EL) spectrum of light in the visible range of 0.38 μm to 0.8 μm. The simulations were run at room temperature (25˚C) using an 832-second time step. A value of 0.038 was observed as the mesh delta of X (μm), Y (μm) and Z (μm). The simulations were run at room temperature (25˚C) using an 832-second time step. A value of 0.038 was observed as the mesh delta of X (μm), Y (μm) and Z (μm). We considered a grid of 100 μm length and 10 μm width, isotropic waveguide, vertical input plane source, Gaussian modulated continuous wave light pulse, Anisotropic perfectly matched layer (APML), and Y-polarization to observe DFT results for conducting the simulation in OptiFDTD simulator. We studied three observation points on or after the outer face of the $SiO_2$ layer for the analysis of the simulation results.

## 4 Results and discussion

We designed various types of white OLEDs structure and changed the physical parameters of white OLEDs by engraving several micro lenses on top of the surface of the substrate layer. Various optical properties of the OLEDs were investigated and analyzed such as EL spectrum i.e. intensity of light, 2D image map and 3D distribution of DFT of Ey, light spreading distribution and EQE of the OLEDs. The performance parameters of basic OLED and MLA engraved OLEDs obtained from simulation results were compared and studied for optimization.

### 4.1 Evaluation of basic OLED's performance

We analyzed the simulation results of the three observation points on or above the outer face of the substrate i.e., OP_1, OP_2 and OP_3 for the basic OLED design of Fig 4(a) considering the light intensity spectrum and light distribution. Fig 5(a)–5(i) demonstrate the simulation results of basic OLED design at the OP_1, OP_2, and OP_3, individually. The variation of EL light intensity of the electric field component in the y-direction (Ey) with regard to wavelength is shown in Fig 5(a), where the highest intensity of 23000 arbitrary units (a.u.) was observed at 620 nm wavelength. Fig 5(b) and 5(c) show the near-field 3D distribution and 2D picture map of DFT of Ey for a basic OLED. Therefore, Fig 5(a)–5(c) confirm the production of white light at the near field with fluctuations in the EL intensity around the top (state of maximum intensity) of the EL spectrum. Figs 5(d), 4(e) and 4(f) show the EL intensity, 3D distribution, and 2D picture map of the DFT of Ey of the basic OLED model in the OP_2. OP_2 had a higher EL intensity of 13500 a.u. at a wavelength of 660 nm, resulting in a smoother and flatter white light distribution than near-field 1. Fig 5(g)–5(i) illustrate the smoother EL intensity spectrum with the focused light beam at OP_2 where the peak EL intensity of 10000 a.u. was obtained at 650 nm. The decrease of maximum EL intensity at OP_1 and OP_2 was caused due to the spreading of light, which is agreed according to the properties of OLED light source.

At OP_2, peak intensity was observed at the red region (650 nm) whereas increasing of light intensity was started from the blue area of the visible spectrum (380 nm to 750 nm). In addition, EL intensity begun to decrease from the red region and confirmed the generation of smooth EL spectrum with white OLED light. Therefore, simulation results confirmed the production of white light in all the observation areas with excellent light spreading for the basic OLED model.

### 4.2 An investigation of the performance of spherical MLA engraved OLEDs

Planoconvex and planoconvex micro lenses (half circles) were etched on the substrate's outer surface to increase light intensity and efficacy, with no space between the lenses. Eight different sized lenses (Spherical-1 to Spherical-8) were considered during the simulation as summarized in Table 2.

**4.2.1 Spherical planoconvex MLA engraved OLEDs.** We completed the simulation with planoconvex spherical MLA engraved OLEDs with Spherical-1 lens having height/radius of 0.4 $\mu$m. The 3D light distribution of DFT of Ey was investigated to observe the light distribution obtained from the OLED at the three observation areas i.e., OP_1, OP_2 and OP_3 on or after the substrate layer of the OLEDs, as demonstrated in Fig 6(a)–6(c).

Afterward, we changed the height/radius of spherical planoconvex lenses as summarized in Table 2. Spherical-2, Spherical-3, and Spherical-4 lenses engraved OLEDs' 3D light dispersion of DFT of Ey at the OP_1, OP_2, and OP_3 are represented in Fig 6(d)–6(l), correspondingly. Excellent consistent light distribution was noted at or after the OP_3 (see Fig 6(c), 6(f), 6(i) and 6(l) although OP_2 also showed better light distribution. In addition, sufficiently consistent light was not identified for the spherical planoconvex lens engraved OLEDs at OP_1. The

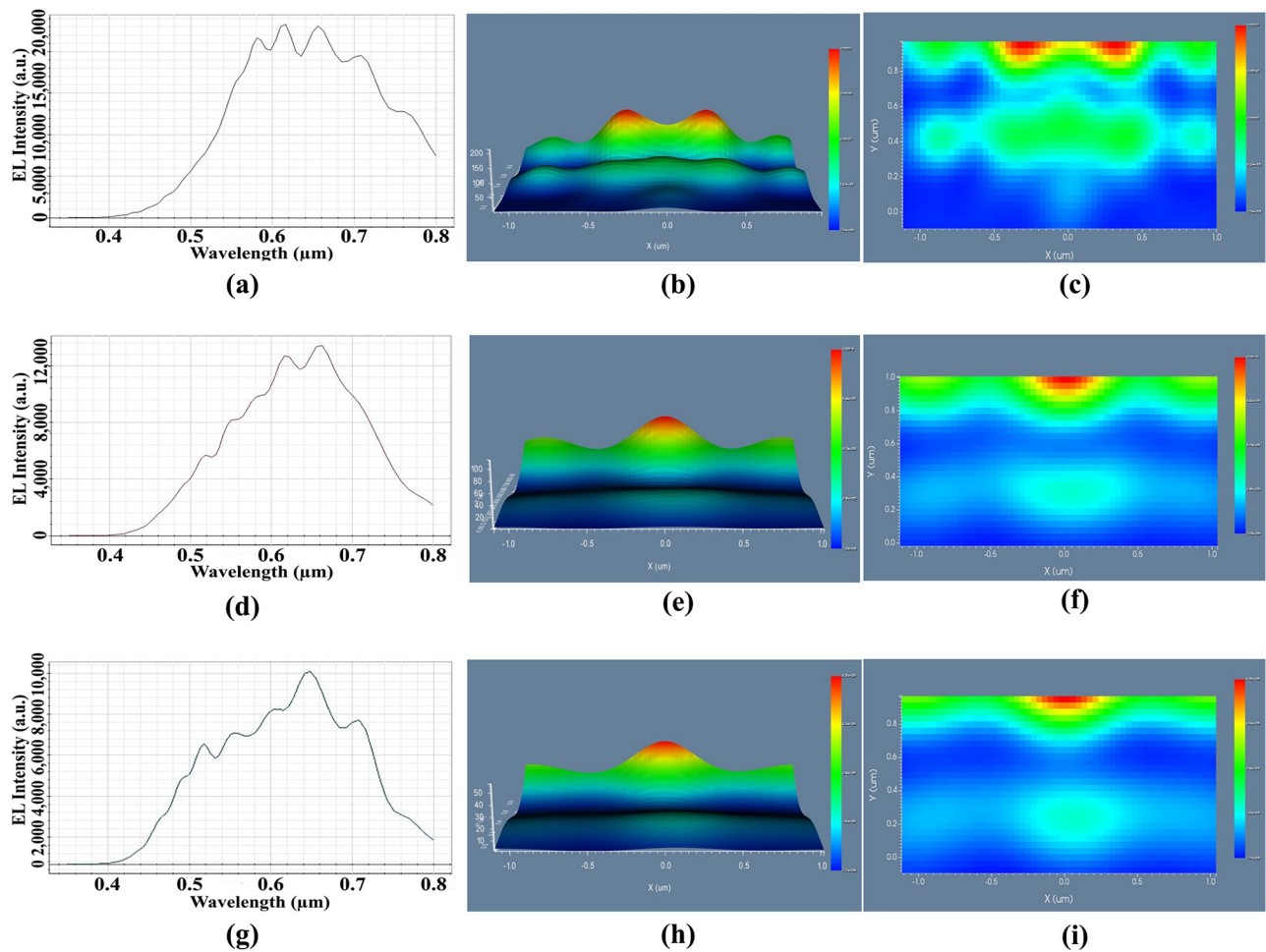

**Fig 5.** Performance evaluation of basic OLED: (a), (d), (g) EL intensity of Ey: (a) OP_1, (d) OP_2, (g) OP_3; (b), (e), (h) 3D light distribution of Ey: (b) OP_1, (e) OP_2, (h) OP_3; (c), (f), (i) 2D image map of Ey: (c) OP_1, (f) OP_2, (i) OP_3.

analysis of 3D light distribution confirmed that Spherical-2 and Spherical-3 showed excellent consistent light distribution among the spherical planoconvex MLA engraved OLEDs (see Fig 6(f) and 6(i)).

Simulation results for 2D image map of DFT of Ey of Spherical-1, Spherical-2, Spherical-3, and Spherical-4 MLA incorporated OLEDs at the OP_1, OP_2 and OP_3 are illustrated in Fig 7(a)–7(l), correspondingly. OP_1 application of spherical planoconvex lenses on the substrate layer's exterior surface caused the OLED light to focus at the lens' focal point, as shown in Fig 7(a), 7(d), 7(g) and 7(j). Smoother light distribution was achieved at the OP_2 as light defocused next to the lens' focal point. Simulation results confirmed that considerable light spreading was observed for all the designed OLEDs in the OP_2 and OP_3 regions. Spherical-2 and Spherical-3 showed better consistent light distribution among the spherical planoconvex MLA engraved OLEDs after OP_3 region (see Fig 7(f) and 7(i)) which are agreed with the 3D light distribution analysis.

Fig 8 shows the EL light intensity of Ey at the OP_3 for the spherical planoconvex MLA etched OLEDs. The detected peak EL light intensity at the OP_3 for Spherical-1, Spherical-2, Spherical-3 and Spherical-4 MLA engraved OLEDs were 30500 a.u. at the wavelength of 610

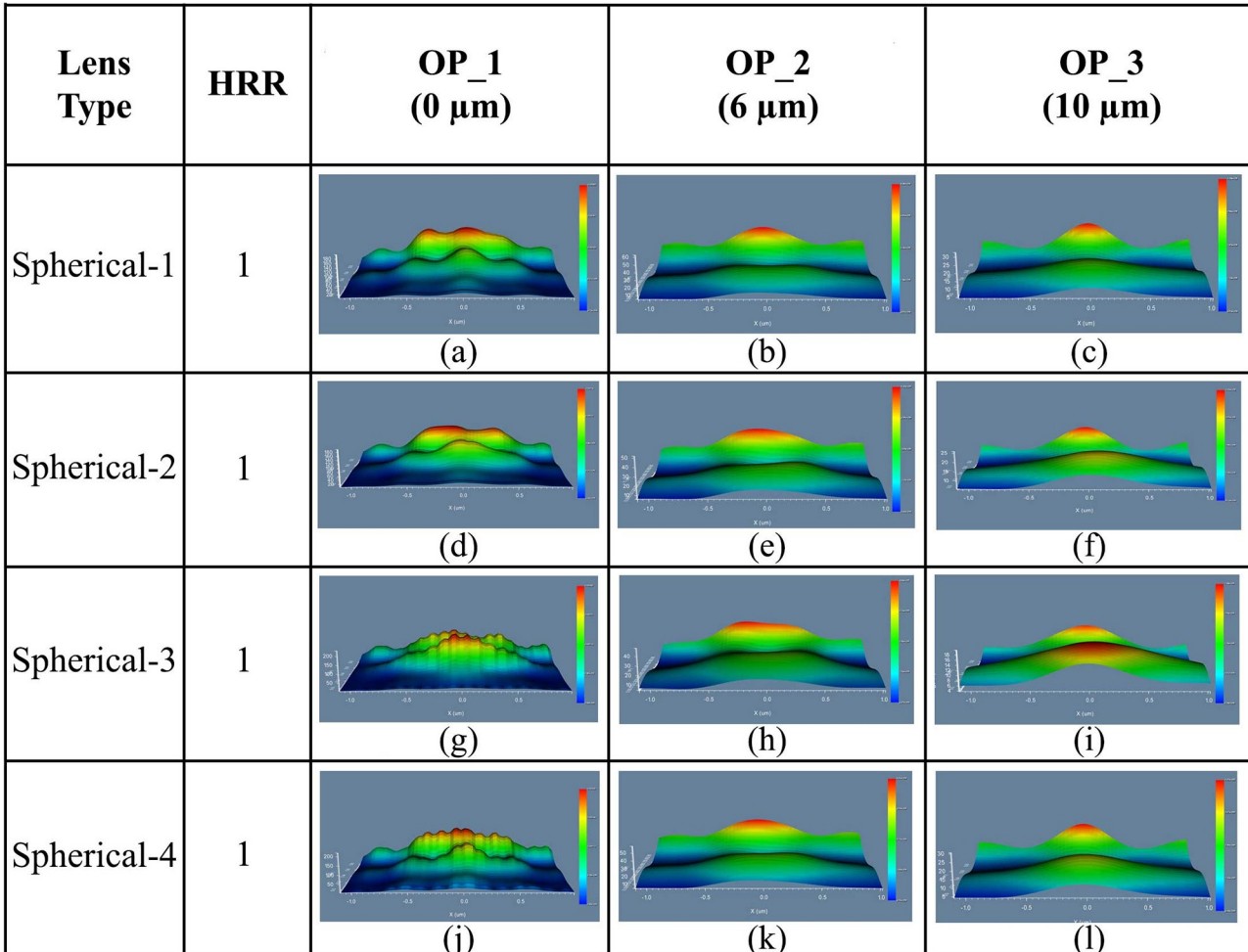

| Lens Type | HRR | OP_1 (0 µm) | OP_2 (6 µm) | OP_3 (10 µm) |
|---|---|---|---|---|
| Spherical-1 | 1 | (a) | (b) | (c) |
| Spherical-2 | 1 | (d) | (e) | (f) |
| Spherical-3 | 1 | (g) | (h) | (i) |
| Spherical-4 | 1 | (j) | (k) | (l) |

**Fig 6.** 3D light distribution of Ey for spherical planoconvex MLA engraved OLEDs: (a)-(c) Spherical-1; (d)-(f) Spherical-2; (g)-(i) Spherical-3; (j)-(l) Spherical 4: (a), (d), (g), (j) Near field; (b), (e), (h), (k) Far field 1; (c), (f), (i), (l) Far field 2.

nm [Fig 8(a)], 33000 a.u. at the wavelength of 615 nm [Fig 8(b)], 35000 a.u. at the wavelength of 615 nm [Fig 8(c)] and 32000 a.u. at the wavelength of 615 nm [Fig 8(d)], respectively. Therefore, all models of spherical planoconvex MLA engraved OLEDs significantly increased the EL light intensity compared to basic OLED model which produced the EL intensity of 10000 arbitrary units [4, 9, 30]. The above phenomena occurred because with the incorporation of spherical planoconvex MLA on the outer surface of OLEDs, curvature shape of the lenses confirms the smaller maximum incident light angles compared to critical angles and led to the light to be refracted rather than reflection. The spherical planoconvex MLA engraved OLEDs (Spherical-1 to Spherical-4) have produced significant light in the visible range which confirmed the generation of white OLED light. Spherical-3 was the optimized lens for the generation of highest EL intensity because an increasing trend was observed up to Spherical-3 with the increase of lens height or radius and afterwards started decreasing the EL light intensity. Therefore, Spherical-3 MLA engraved OLED can be considered as the best OLED structure among the spherical planoconvex MLA engraved OLEDs.

**4.2.2 Spherical planoconcave MLA engraved OLEDs.** The 3D light distribution of DFT of Ey for spherical planoconcave lenses i.e. Spherical-5, Spherical-6, Spherical-7, and

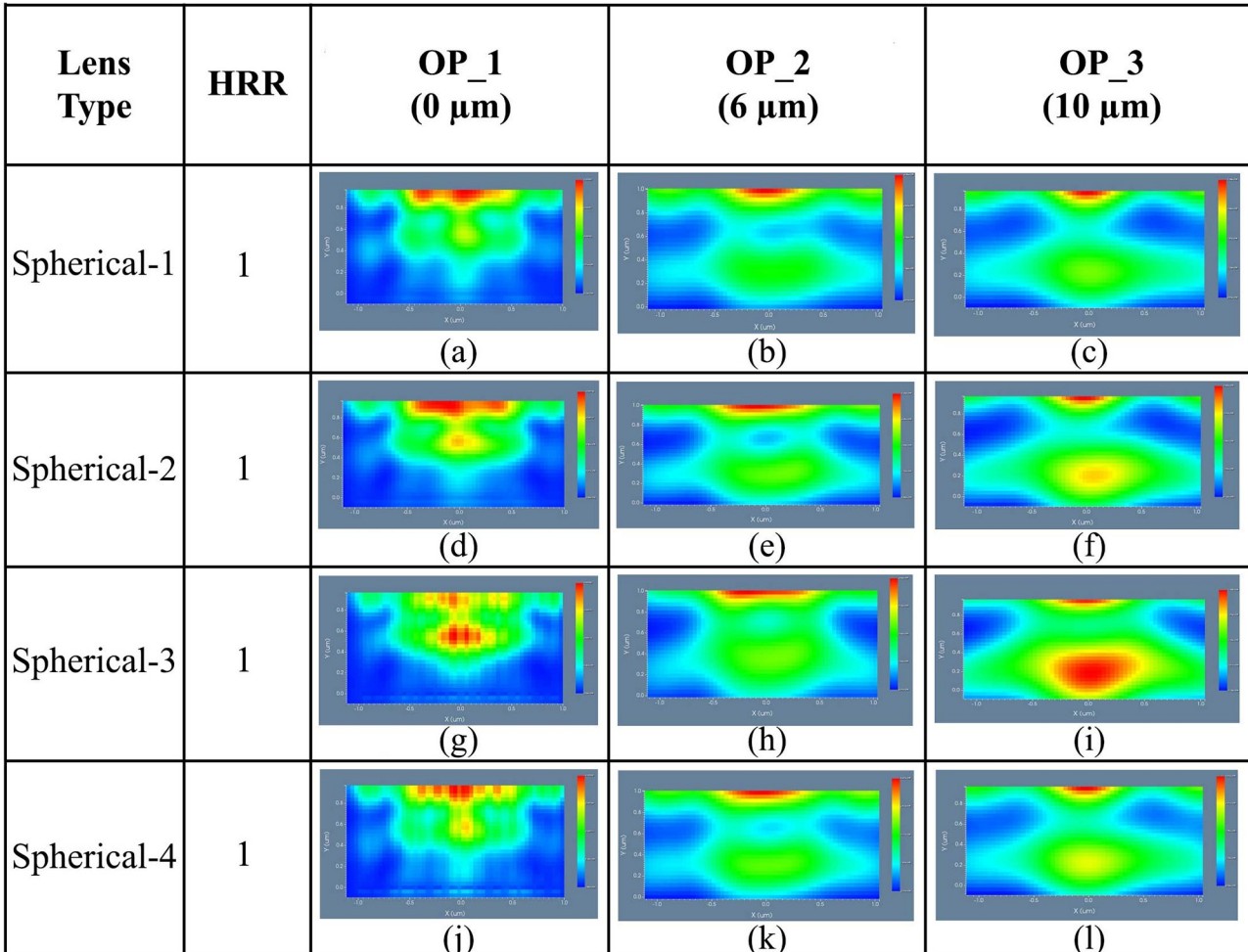

| Lens Type | HRR | OP_1 (0 μm) | OP_2 (6 μm) | OP_3 (10 μm) |
|---|---|---|---|---|
| Spherical-1 | 1 | (a) | (b) | (c) |
| Spherical-2 | 1 | (d) | (e) | (f) |
| Spherical-3 | 1 | (g) | (h) | (i) |
| Spherical-4 | 1 | (j) | (k) | (l) |

**Fig 7.** 2D image map of Ey for spherical planoconvex MLA engraved OLEDs: (a)-(c) Spherical-1; (d)-(f) Spherical-2; (g)-(i) Spherical-3; (j)-(l) Spherical 4: (a), (d), (g), (j) OP_1; (b), (e), (h), (k) OP_2; (c), (f), (i), (l) OP_3.

Spherical-8 MLA incorporated OLEDs at the OP_1, OP_2 and OP_3 are demonstrated in Fig 9(a)–9(l), respectively. Consistent light distribution was evident for all the spherical planoconcave MLA engraved OLEDs after the OP_2 and OP_3 although light distribution was not consistent sufficiently at the OP_1. In addition, excellent light distribution was seen for spherical planoconcave MLA engraved OLEDs after OP_3 which can be justified from Fig 9(c), 9(f), 9(i) and 9(l). A closer observation confirmed that Spherical-7 MLA engraved OLED described excellent light distribution among the spherical planoconcave MLA engraved OLEDs (see Fig 9(i)). We also investigated the 2D image map for of DFT of Ey for spherical MLA engraved OLEDs at OP_1, OP_2 and OP_3, as illustrated in Fig 10. Emitted light started spreading after OP_1 because planoconcave lenses were incorporated on the substrate layer, as shown in Fig 10(a), 10(d), 10(g) and 10(j). The light spreading was improved as it travels along the observation regions i.e. OP_2 and OP_3. Therefore, excellent light spreading was observed at OP_3, as demonstrated in Fig 10(c), 10(f), 10(i) and 10(l) compared to the light spreading at OP_2 and OP_1.

In addition, Spherical-7 MLA engraved OLED indicated excellent light spreading among the spherical planoconcave MLA engraved OLEDs (see Fig 10(i)), which is justified with the

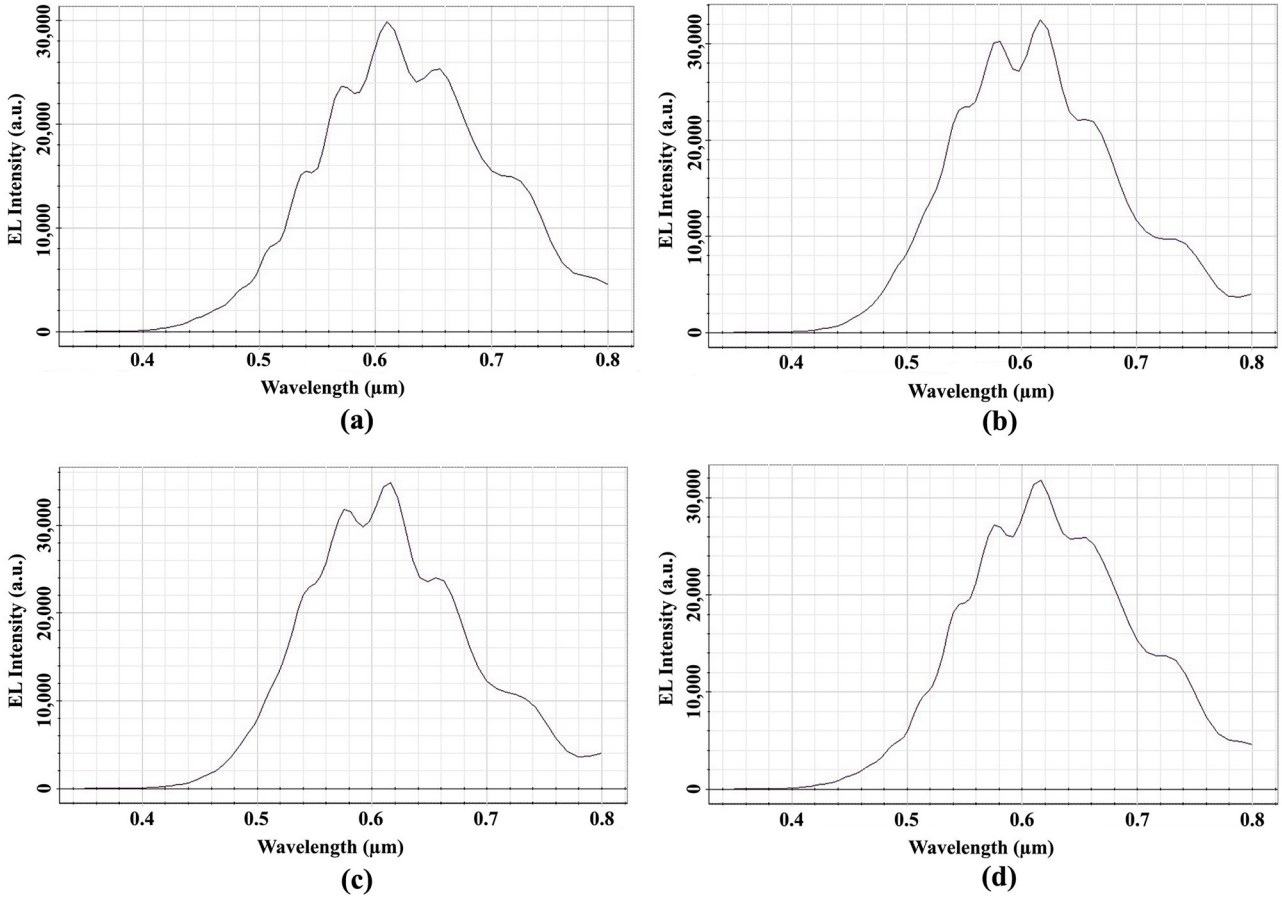

**Fig 8.** EL Intensity of Ey for spherical planoconvex MLA engraved OLEDs: (a) Spherical-1; (b) Spherical-2; (c) Spherical-3; (d) Spherical 4.

3D light distribution. Overall, spherical planoconcave MLA engraved OLEDs showed more light spreading compared to spherical planoconvex MLA engraved OLEDs in all the models.

OP_2 Ey light intensity is shown in Fig 11 for circular planoconcave MLA etched OLEDs in the OP_3. The peak EL intensity of 28000 a.u. was detected at a wavelength of 610 nm for Spherical-5 MLA engraved OLED at OP_3, as shown in Fig 11(a). For Spherical-2, Spherical-3 and Spherical-4 MLA engraved OLEDs, the peak EL light intensity of 31000 a.u. at the wavelength of 610 nm [Fig 11(b)], 36000 a.u. at the wavelength of 615 nm [Fig 11(c)] and 32200 a.u. at the wavelength of 615 nm [Fig 11(d)] were obtained at OP_3, respectively. As a result, all the spherical planoconcave MLA engraved OLEDs significantly enhanced the EL light intensity compared to the basic OLED model.

In addition, white OLED light was guaranteed from all the spherical planoconcave MLA engraved OLEDs as there were significant light in the visible spectrum. The EL intensity showed an increasing trend up to Spherical-7 with the increase of lens height or radius started from Spherical-5 and afterwards started decreasing the EL light intensity. Therefore, Spherical-7 was the optimized lens for the generation of highest EL intensity among spherical planoconcave MLA engraved OLEDs. Thus, 0.6 $\mu$m lens' radius/height can be judged as the optimized radius/height for spherical lenses as Spherical-3 and Spherical-7 both have the lens radius/height of 0.6 $\mu$m. Additionally, Spherical-7 MLA engraved OLED produced the higher peak EL intensity compared to Spherical-3 MLA engraved OLED. Consequently, Spherical-7

| Lens Type | HRR | OP_1 (0 μm) | OP_2 (6 μm) | OP_3 (10 μm) |
|---|---|---|---|---|
| Spherical-5 | 1 | (a) | (b) | (c) |
| Spherical-6 | 1 | (d) | (e) | (f) |
| Spherical-7 | 1 | (g) | (h) | (i) |
| Spherical-8 | 1 | (j) | (k) | (l) |

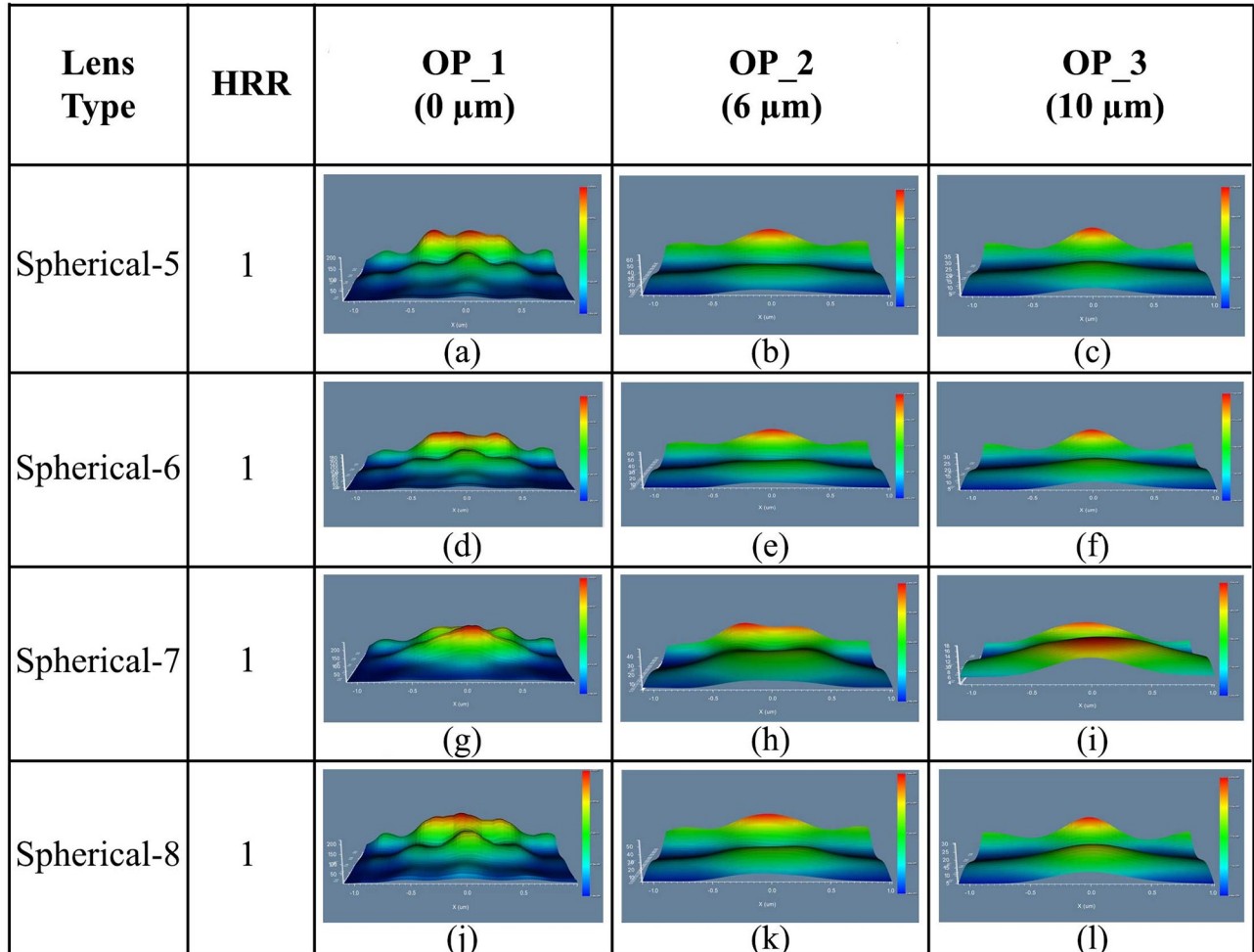

**Fig 9.** 3D light distribution of Ey for spherical planoconcave MLA engraved OLEDs: (a)-(c) Spherical-5; (d)-(f) Spherical-6; (g)-(i) Spherical-7; (j)-(l) Spherical 8: (a), (d), (g), (j) OP_1; (b), (e), (h), (k) OP_2; (c), (f), (i), (l) OP_3.

type MLA engraved OLED can be considered as the best OLED among the designed spherical MLA (both planoconvex and planoconcave) incorporated OLEDs.

### 4.3 Elliptical MLA engraved OLED performance evaluation

OLEDs' performance was also evaluated with the help of elliptical lenses. Different types of elliptical lenses (Concave-1 to Concave-5 and Convex-1 to Convex-5) were considered for the performance evaluation of OLEDs during the simulation as summarized in Table 2. The planned OLEDs' elliptical lenses were incorporated on the exterior surface of the substrate layer.

**4.3.1 Elliptical planoconvex MLA engraved OLEDs.** The simulation was started with elliptical planoconvex MLA engraved OLED for Convex-1 lens having a HRR of 1.25. The 3D light distribution of DFT of Ey for Convex-1 MLA engraved OLED at the OP_1, OP_2 and OP_3 are depicted in Fig 12(a)–12(c). The 3D light distribution of DFT of Ey for Convex-2, Convex-3, Convex-4 and Convex-5 MLA engraved OLEDs at the OP_1, OP_2 and OP_3 are illustrated in Fig 12(d)–12(o), correspondingly.

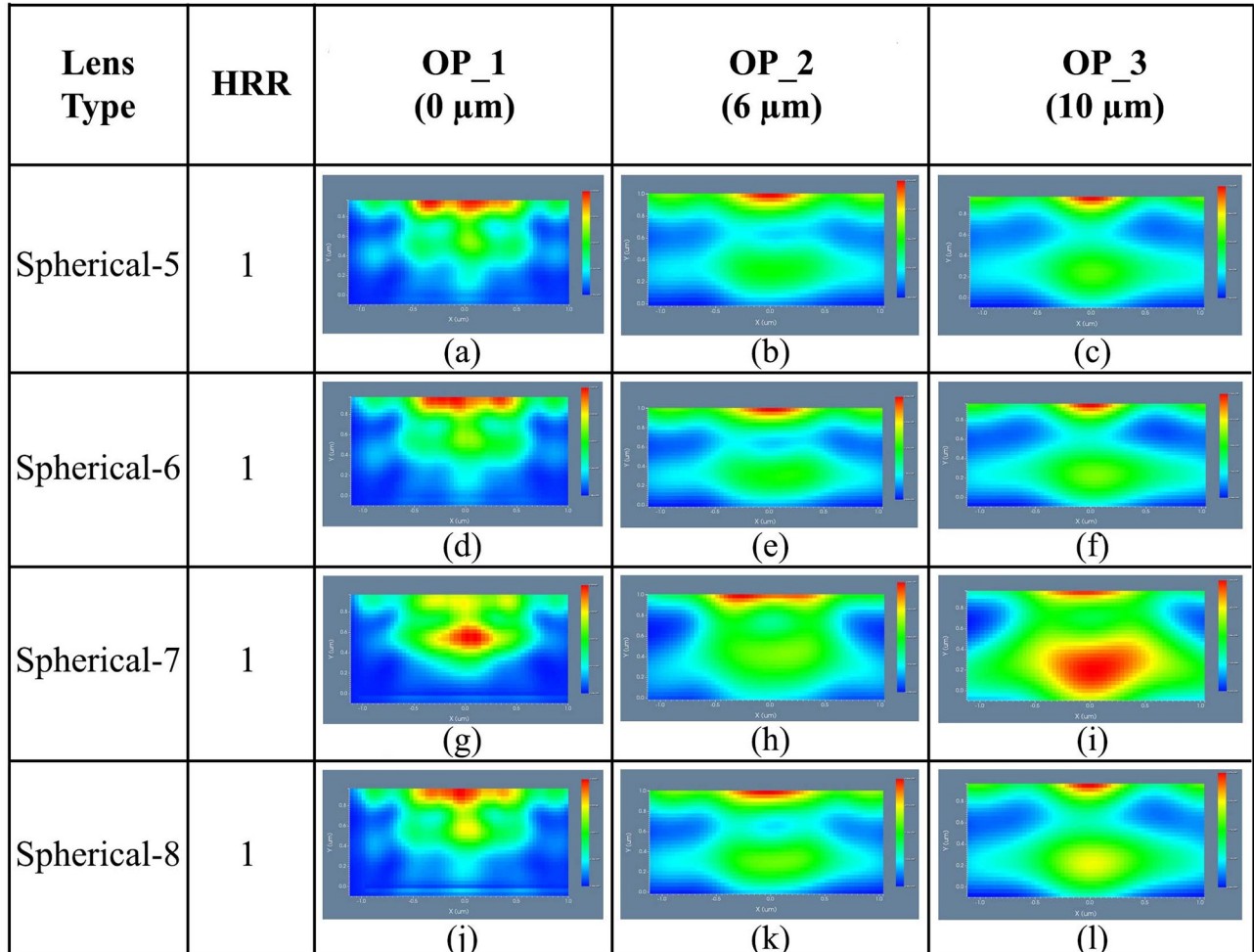

| Lens Type | HRR | OP_1 (0 μm) | OP_2 (6 μm) | OP_3 (10 μm) |
|---|---|---|---|---|
| Spherical-5 | 1 | (a) | (b) | (c) |
| Spherical-6 | 1 | (d) | (e) | (f) |
| Spherical-7 | 1 | (g) | (h) | (i) |
| Spherical-8 | 1 | (j) | (k) | (l) |

**Fig 10.** 2D image map of Ey for spherical planoconcave MLA engraved OLEDs: (a)-(c) Spherical-5; (d)-(f) Spherical-6; (g)-(i) Spherical-7; (j)-(l) Spherical 8: (a), (d), (g), (j) OP_1; (b), (e), (h), (k) OP_2; (c), (f), (i), (l) OP_3.

Though light distribution was not consistent at OP_1 for all the lenses, it was increased significantly at the OP_2 and OP_3 locations. Excellent light distribution was evident at the OP_3 for all types of elliptical planoconvex MLA engraved OLEDs where Convex-1 type lens engraved OLED showed best simulation result.

Fig 13 demonstrated the 2D image map of DFT of Ey for the designed elliptical planoconvex MLA engraved OLEDs showing the light spreading profile at the different observation areas. Generated light was focused on the lenses' focal points at the near field location, as illustrated in Fig 13(a), 13(d), 13(g), 13(j) and 13(m) because planoconvex lenses were considered for simulation.

After the focal point location, light started to spread consistently and continued up to OP_2 and OP_3, as illustrated in [Fig 13(b), 13(e), 13(h), 13(k), 13(n), 13(c), 13(f), 13(i), 13(l) and 13(o)], respectively. Consistent light spreading was evident from the 2D maps distribution for all the elliptical planoconvex MLA engraved OLEDs, where Convex-1 type lens engraved OLED indicated best simulation result. In addition, significant enhancement in light intensity profile was confirmed in the elliptical planoconvex MLA engraved OLEDS, compared to spherical planoconvex MLA engraved OLEDs.

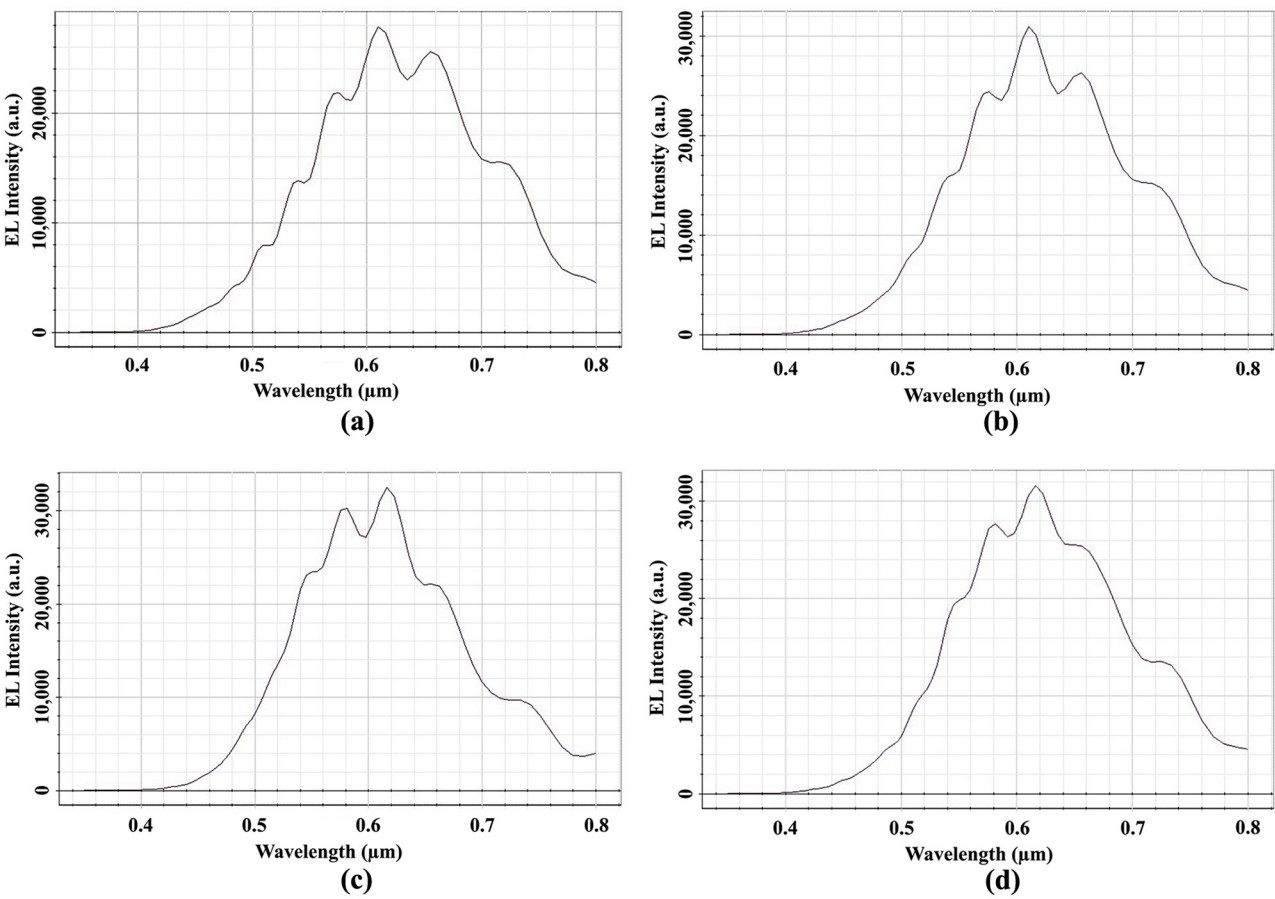

**Fig 11.** EL Intensity of Ey for spherical planoconcave MLA engraved OLEDs: (a) Spherical-5; (b) Spherical-6; (c) Spherical-7; (d) Spherical 8.

The EL light intensity of Ey at the distant OP_3 for the elliptical planoconvex MLA etched OLEDs is shown in Fig 14. The highest EL intensity of 38000 a.u. was identified at a wavelength of 680 nm for Convex-1 MLA engraved OLED at OP_3, as depicted in Fig 14(a). For Convex-2, Convex-3, Convex -4 and Convex-5 MLA incorporated OLEDs, the peak EL light intensity of 27000 a.u. at 675 nm wavelength [Fig 14(b)], 31400 a.u. at 680 nm wavelength [Fig 14(c)], 35000 a.u. at 690 nm wavelength [Fig 14(d)] and 37000 a.u. at 720 nm wavelength [Fig 14(e)] were observed at OP_3, separately. Therefore, all the models of elliptical planoconvex MLA engraved OLEDs significantly increased the EL light intensity compared to basic OLED model.

Considerable amount of light was observed in the visible range which confirmed the production of white OLEDs for all the designed models of elliptical planoconvex MLA engraved OLEDs. Convex-1 MLA engraved OLED showed higher light intensity among the elliptical planoconvex MLA engraved OLEDs with consistent light distribution.

**4.3.2 Elliptical planoconcave MLA engraved OLEDs.** We investigated the performance evaluation for five types (Concave-1 to Concave-5) of elliptical planoconcave MLA engraved OLEDs. The 3D light distribution of DFT of Ey for elliptical planoconcave MLA engraved OLEDs at the OP_1, OP_2 and OP_3 are depicted in Fig 15. Inconsistent light distribution was found in OP_1 for all the models, as shown in Fig 15(a), 15(d), 15(g), 15(j) and 15(m).

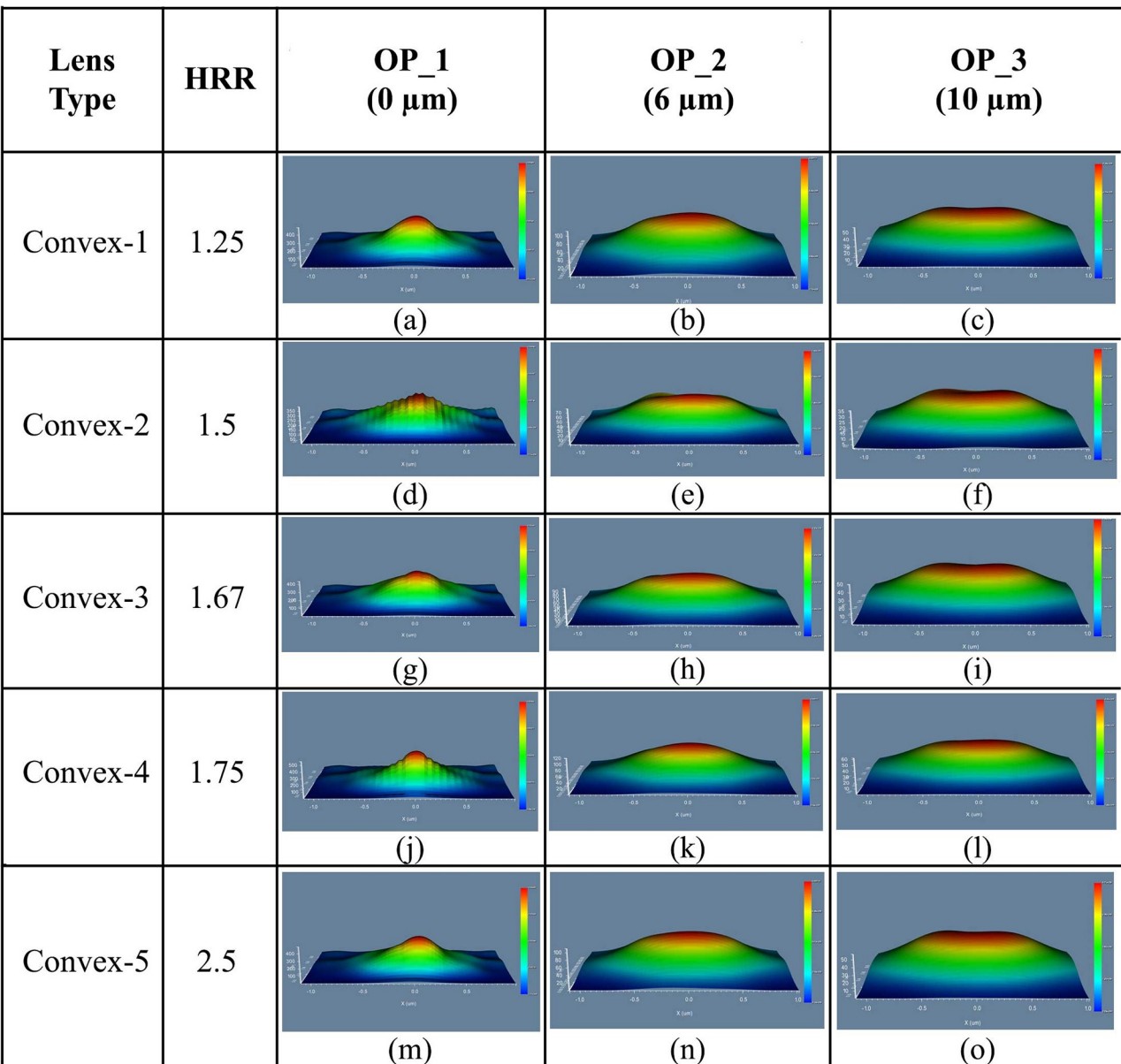

| Lens Type | HRR | OP_1 (0 μm) | OP_2 (6 μm) | OP_3 (10 μm) |
|---|---|---|---|---|
| Convex-1 | 1.25 | (a) | (b) | (c) |
| Convex-2 | 1.5 | (d) | (e) | (f) |
| Convex-3 | 1.67 | (g) | (h) | (i) |
| Convex-4 | 1.75 | (j) | (k) | (l) |
| Convex-5 | 2.5 | (m) | (n) | (o) |

**Fig 12.** 3D light distribution of Ey for elliptical planoconvex MLA engraved OLEDs: (a)-(c) Convex-1; (d)-(f) Convex-2; (g)-(i) Convex-3; (j)-(l) Convex-4; (m)-(o) Convex-5: (a), (d), (g), (j), (m) OP_1; (b), (e), (h), (k), (n) OP_2; (c), (f), (i), (l), (o) OP_3.

In the OP_2 and OP_3 areas, light distribution was improved by increasing the distance from the substrate layer, which was apparent from the 3D light distribution at OP_2 [Fig 15 (b), 15(e), 15(h), 15(k) and 15(n)] and OP_3 [Fig 15(c), 15(f), 15(i), 15(l) and 15(o)]. Therefore, excellent light distribution was noticed at the OP_3 compared to OP_2 and OP_1 for all the elliptical planoconcave MLA engraved OLEDs, where Concave-5 MLA engraved OLED showed best 3D light distribution. Fig 16 illustrates the 2D image map for elliptical planoconcave MLA engraved OLEDs (Concave-1 to Concave-5) taking light spreading into account.

Due to the concave nature (focal point effect) of the lenses, inconsistent light spreading was noticed at the OP_1, as shown in Fig 16(a), 16(d), 16(g), 16(j) and 16(m). The light spreading

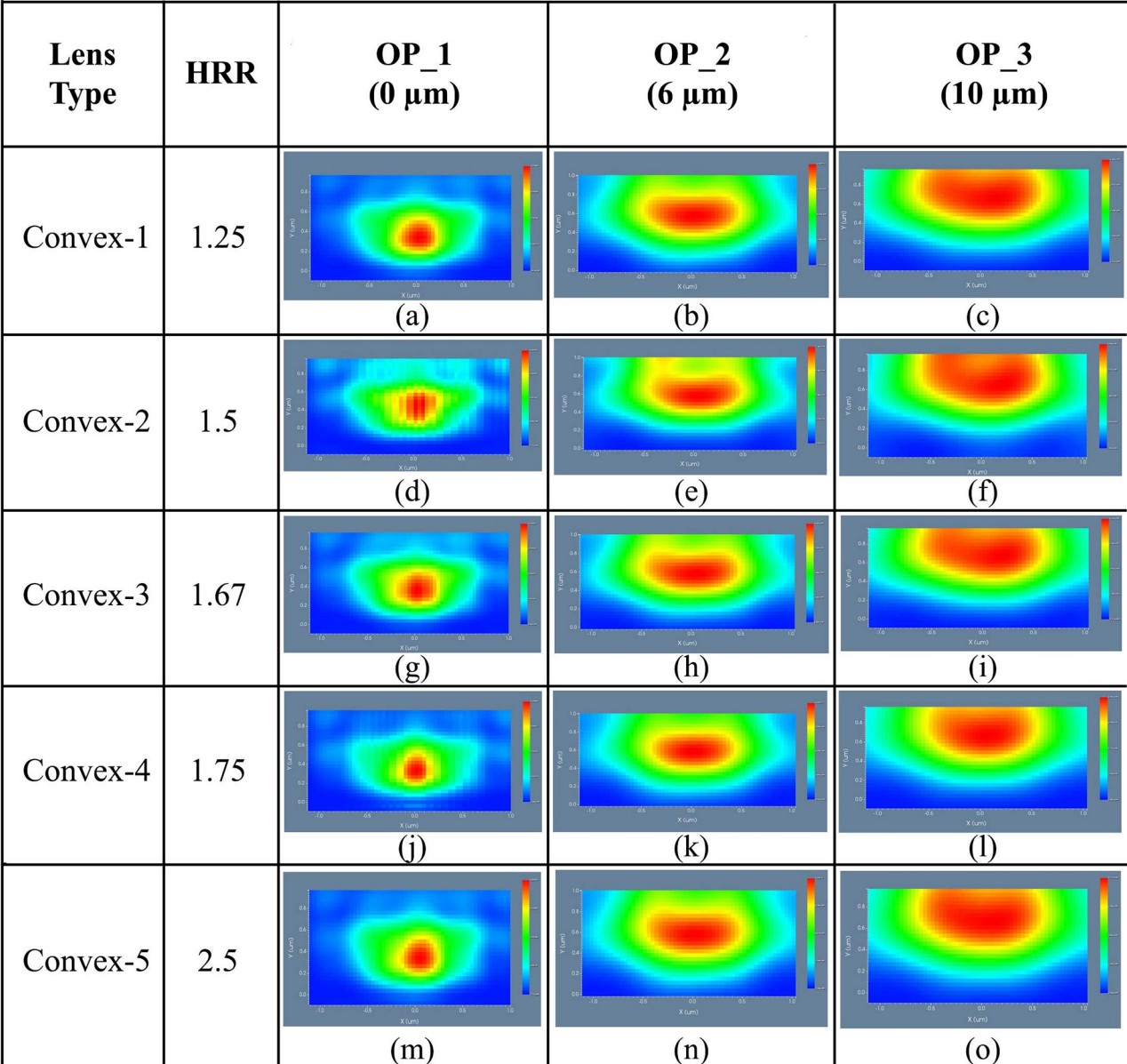

| Lens Type | HRR | OP_1 (0 μm) | OP_2 (6 μm) | OP_3 (10 μm) |
|-----------|-----|-------------|-------------|--------------|
| Convex-1 | 1.25 | (a) | (b) | (c) |
| Convex-2 | 1.5 | (d) | (e) | (f) |
| Convex-3 | 1.67 | (g) | (h) | (i) |
| Convex-4 | 1.75 | (j) | (k) | (l) |
| Convex-5 | 2.5 | (m) | (n) | (o) |

**Fig 13.** 2D image map of Ey for elliptical planoconvex MLA engraved OLEDs: (a)-(c) Convex-1; (d)-(f) Convex-2; (g)-(i) Convex-3; (j)-(l) Convex-4; (m)-(o) Convex-5: (a), (d), (g), (j), (m) OP_1; (b), (e), (h), (k), (n) OP_2; (c), (f), (i), (l), (o) OP_3.

was improved at the OP_2 [Fig 16(b), 16(e), 16(h), 16(k) and 16(n)] and OP_3 [Fig 16(c), 16 (f), 16(i), 16(l) and 16(o)] for all the OLED models as those areas were located after the focal point distance of the lenses. Therefore, excellent light spreading was observed at OP_3 for all the elliptical planoconcave MLA engraved OLEDs, whereas Concave-5 MLA engraved OLED showed the best simulation results considering light spreading capability.

Fig 17 shows the EL light intensity of Ey at the OP_3 for the elliptical planoconcave MLA engraved OLEDs.

The peak EL intensity of 31200 a.u. was detected at a wavelength of 685 nm for Concave-1 MLA incorporated OLED at OP_3, as depicted in Fig 17(a). The peak EL light intensity of 33000 a.u. at 685 nm wavelength [Fig 17(b)], 30000 a.u. at 685 nm wavelength [Fig 17(c)],

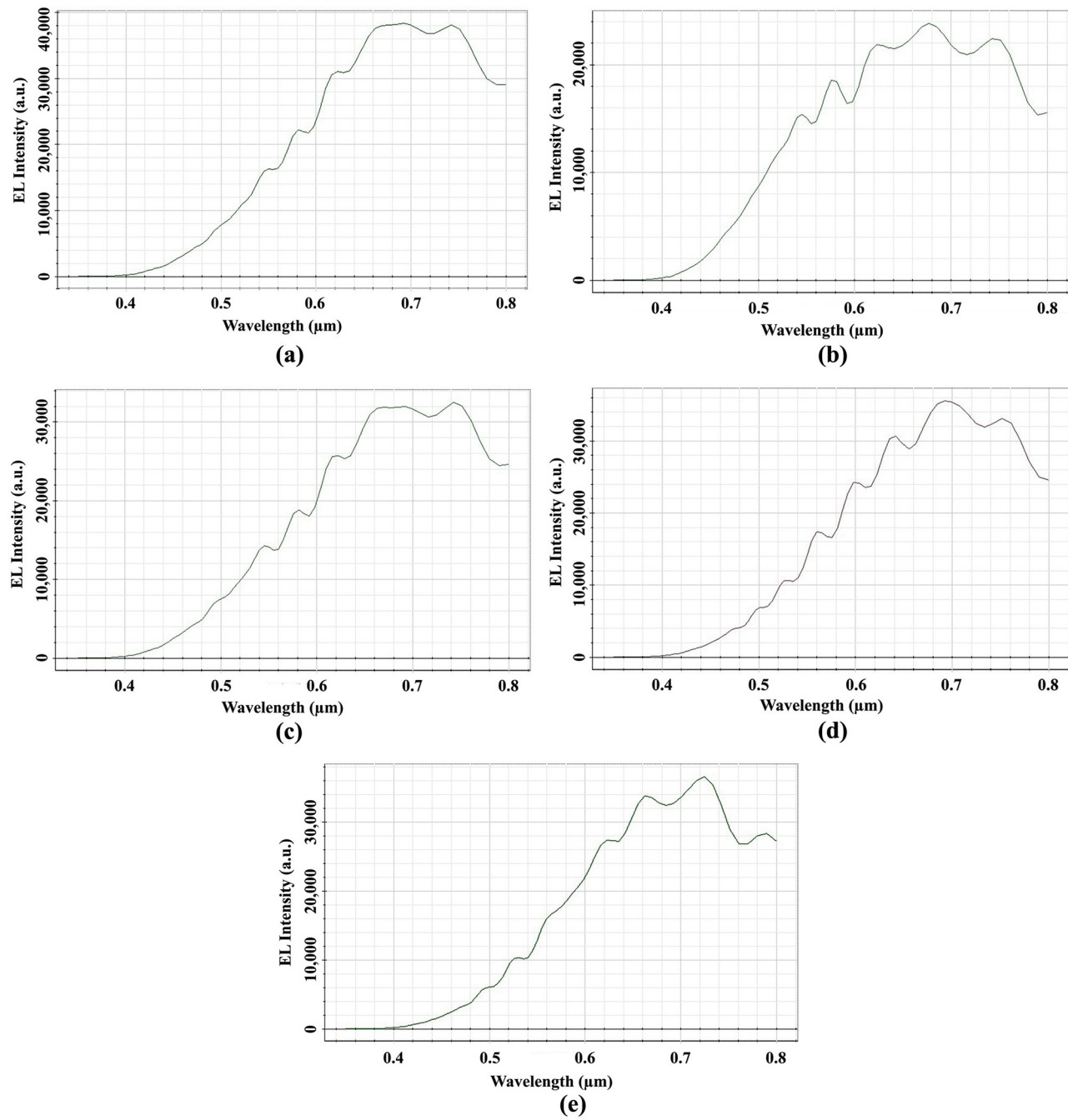

**Fig 14.** EL Intensity of Ey for elliptical planoconvex MLA engraved OLEDs: (a) Convex-1; (b) Convex-2; (c) Convex-3; (d) Convex-4; (e) Convex-5.

30900 a.u. at 735 nm wavelength [Fig 17(d)] and 36000 a.u. at 720 nm wavelength [Fig 17(e)] were detected at OP_3 for Concave-2, Concave-3, Concave-4 and Concave-5 MLA incorporated OLEDs. The EL light intensity was significantly increased for all the models of elliptical planoconvex MLA engraved OLEDs compared to basic OLED model. In addition, all the models of elliptical planoconvex MLA structures formed the white OLEDs as significant quantity of light were detected in the visible spectrum. Among all the elliptical planoconvex MLA

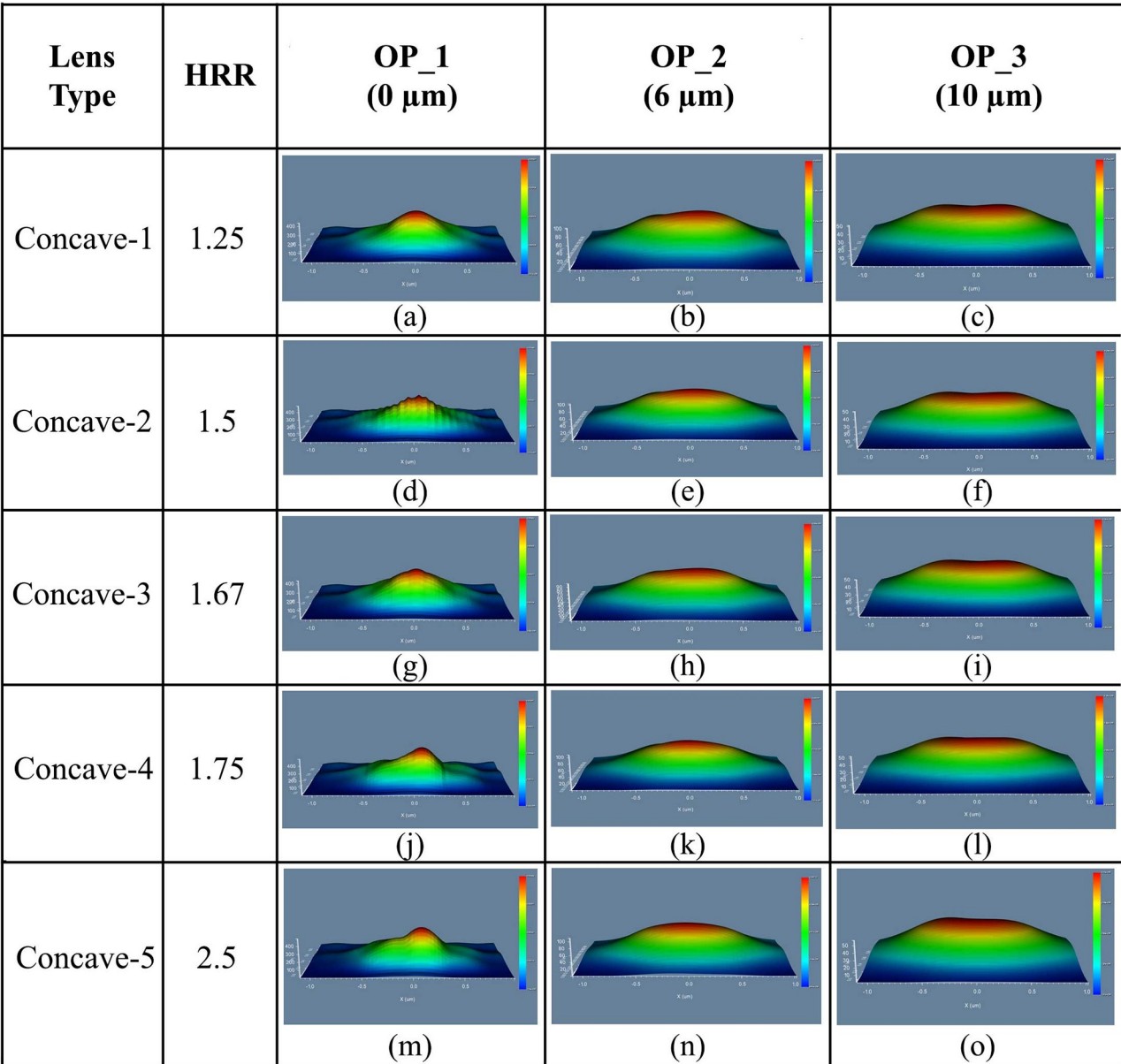

**Fig 15.** E3D light distribution of Ey for elliptical planoconcave MLA engraved OLEDs: (a)-(c) Concave-1; (d)-(f) Concave-2; (g)-(i) Concave-3; (j)-(l) Concave-4; (m)-(o) Concave-5: (a), (d), (g), (j), (m) OP_1; (b), (e), (h), (k), (n) OP_2; (c), (f), (i), (l), (o) OP_3.

engraved OLEDs, Concave-5 MLA engraved OLED showed higher light intensity having excellent light spreading. Convex-1 MLA engraved OLED outperforms Concave-5 MLA engraved OLED considering peak EL light intensity at OP_3 among all types of spherical and elliptical lenses incorporated OLEDs.

## 4.4 Performance evaluation of MLA engraved OLEDs based on physical parameters

We considered EQE and light distribution with intensity for the performance evaluation of the OLEDs, which are very important parameters for OLEDs. EQE of an OLED structure can be

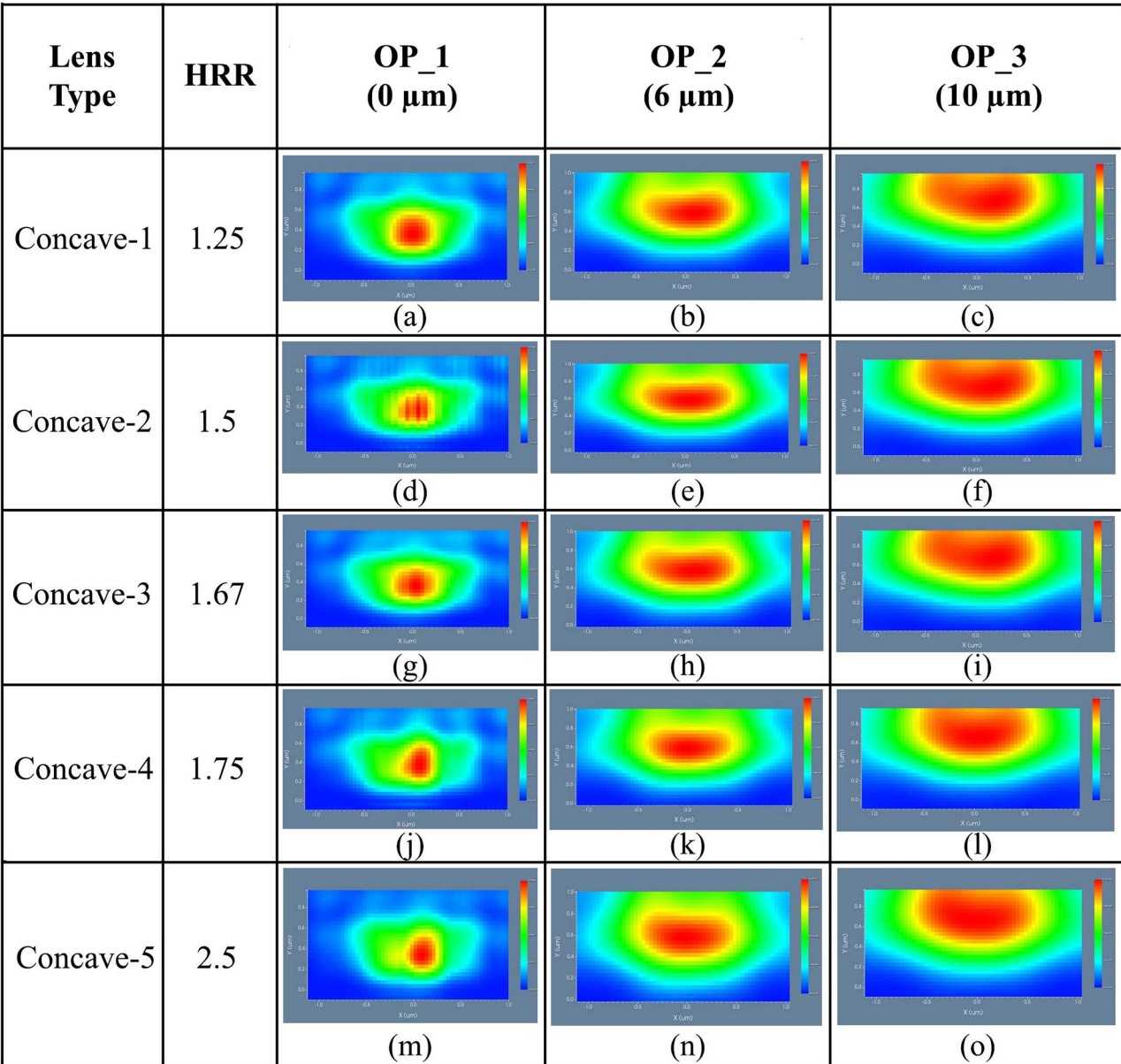

| Lens Type | HRR | OP_1 (0 μm) | OP_2 (6 μm) | OP_3 (10 μm) |
|---|---|---|---|---|
| Concave-1 | 1.25 | (a) | (b) | (c) |
| Concave-2 | 1.5 | (d) | (e) | (f) |
| Concave-3 | 1.67 | (g) | (h) | (i) |
| Concave-4 | 1.75 | (j) | (k) | (l) |
| Concave-5 | 2.5 | (m) | (n) | (o) |

**Fig 16.** 2D image map of Ey for elliptical planoconcave MLA engraved OLEDs: (a)-(c) Concave-1; (d)-(f) Concave-2; (g)-(i) Concave-3; (j)-(l) Concave-4; (m)-(o) Concave-5: (a), (d), (g), (j), (m) OP_1; (b), (e), (h), (k), (n) OP_2; (c), (f), (i), (l), (o) OP_3.

obtained from effective refractive index (ERI). We determined ERI for the designed OLEDs structure during simulation and calculated the EQE for the OLEDs using Eq 3. EQE showed inversely proportional relationship with ERI. The EQE of the basic OLED model was 14.45% for an ERI of 1.86. The EQE of the spherical planoconvex MLA engraved OLEDs i.e. Spherical-1, Spherical-2, Spherical-3 and Spherical-4 were 15.43%, 16.51%, 16.71% and 15.96% for the ERI of 1.80, 1.74, 1.73, 1.77, correspondingly. Therefore, Spherical-3 (0.6 μm radius/height) MLA engraved OLED achieved the highest EQE among all the spherical planoconvex MLA engraved OLEDs. On the contrary, spherical planoconcave MLA engraved OLEDs such as Spherical-5, Spherical-6, Spherical-7 and Spherical-8 achieved the EQE of 15.09%, 15.61%,

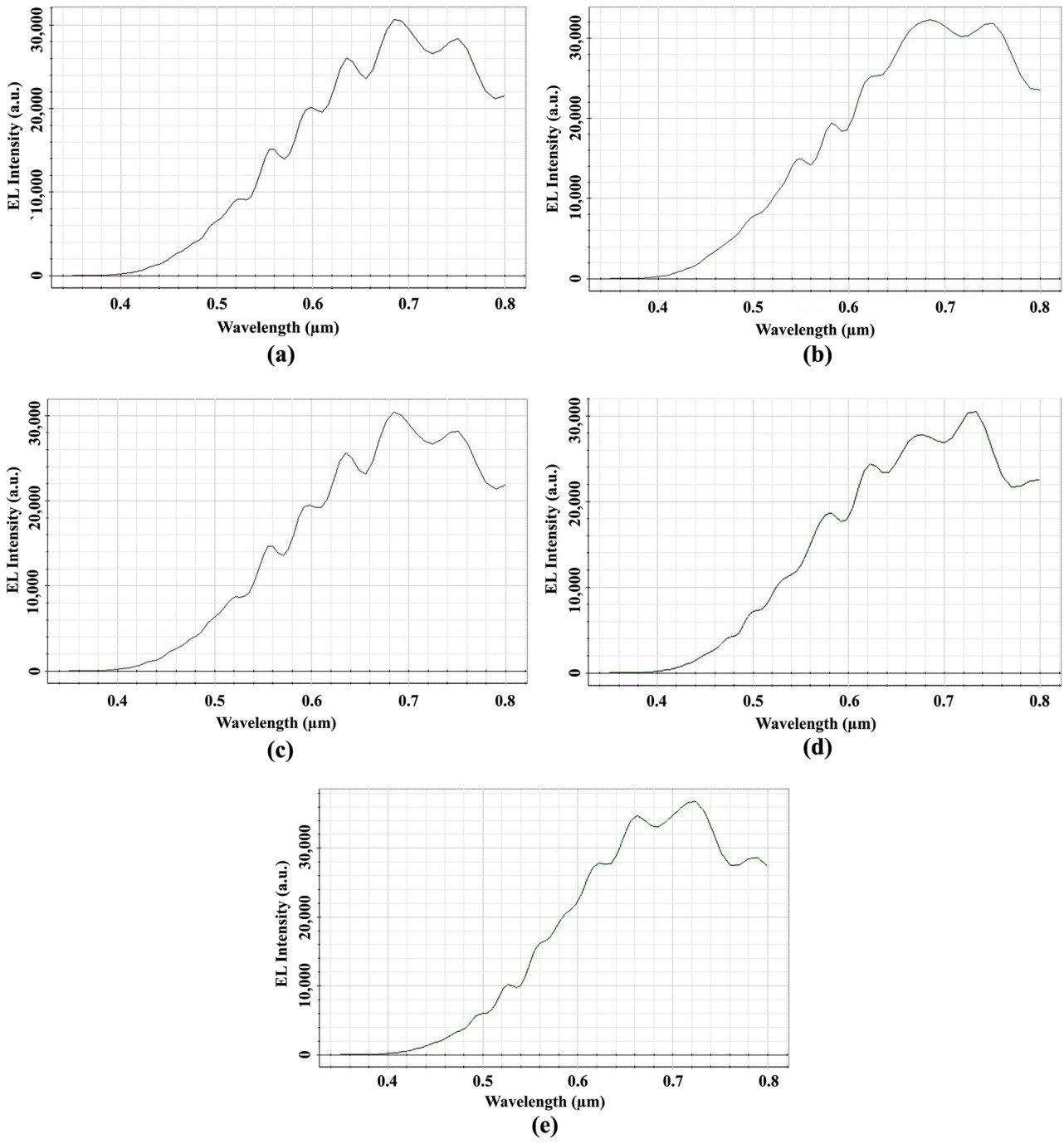

**Fig 17.** EL Intensity of Ey for elliptical planoconcave MLA engraved OLEDs: (a) Concave-1; (b) Concave-2; (c) Concave-3; (d) Concave-4; (e) Concave-5.

16.80% and 16.14% for the ERI of 1.82, 1.79, 1.725 and 1.76, respectively. Consequently, highest EQE was obtained for Spherical-7 (0.6 $\mu$m radius/height) MLA engraved OLED among all the spherical planoconcave MLA engraved OLEDs. In addition, spherical planoconcave (Spherical-7) MLA engraved OLED achieved the better EQE compared to spherical planoconvex MLA engraved OLEDs. Additionally, 0.6 $\mu$m radius/height can be considered as the

optimized value for spherical lenses, as both Spherical-3 and Spherical-7 have the lens radius/ height of 0.6 $\mu$m. The EQE of the Convex-1, Convex-2, Convex-3, Convex-4 and Convex-5 (elliptical planoconvex lenses) MLA engraved OLEDs were 17.30%, 14.93%, 15.69%, 16.71% and 17.10% for the ERI of 1.7, 1.83, 1.785, 1.73 and 1.71, in that order. Thus, Convex-1 (h: 0.5 $\mu$m and r: 0.4 $\mu$m) MLA engraved OLED achieved the highest EQE among the elliptical plano-convex MLA engraved OLEDs with the lowest HRR of 1.25. On the contrast, elliptical plano-concave MLA engraved OLEDs such as Concave-1, Concave-2, Concave-3, Concave-4 and Concave-5 obtained the EQE of 15.78%, 16.33%, 15.26%, 15.52% and 16.90% for the ERI of 1.78, 1.75, 1.81, 1.795 and 1.72, correspondingly. Hence, among the elliptical planoconcave MLA engraved OLEDs, Concave-5 (h: 0.5 $\mu$m and r: 0.2 $\mu$m) MLA engraved OLED showed the highest EQE with the highest HRR of 2.5, opposite to elliptical planoconvex design. Addi-tionally, elliptical planoconvex (Convex-1) MLA engraved OLED achieved the better EQE compared to elliptical planoconcave MLA engraved OLEDs. From the above discussion, it was confirmed that Convex-1 elliptical planoconvex MLA incorporated OLED obtained the high-est EQE of 17.30% for the lowest ERI of 1.7, which achieved a higher EQE of 2.85% compared to the basic OLED. The EQE with the corresponding ERI is illustrated in Fig 18 for all the designed OLEDs including basic OLED. Simulated data show that MLA on the outer surface of the substrate layer considerably enhances the EL light intensity in the visible range and EQE of the developed OLEDs, and validates the generation of white light.

Nevertheless, all kinds of designed OLEDs are not appropriate for commercial applications because consistent light spreading and distribution are required for the production of com-mercial OLEDs. Therefore, it is important to select the proper OLED structure for commercial applications. The addition of MLA on the substrate layer facilitates the emitted lights to inter-fere with one another and generate constructive patterns and destructive patterns in different locations. The constructive and destructive patterns and distance between them determine the light spreading and smoothness of the OLEDs. The shape and nature of the micro lens array have an impact on these patterns. Although, the smoothness of light can be increased with the constructive interference pattern which is located close to one another, but any direct relation-ship between physical parameters of MLA and light distribution is not feasible. As mentioned before, elliptical MLA (Convex-1) engraved OLEDs showed higher peak EL intensity and EQE compared to spherical MLA (Spherical-7) engraved OLEDs, which was caused because of lens aberration effect [32].

The generated light of the OLED reflected back into the OLED device from the air/substrate interface because of total internal reflection (TIR). This phenomenon occurred since air has lower refractive index compared to substrate. Therefore, micro lens array is engraved on the substrate layer to reduce the TIR. OLED light extraction efficiency can be improved and the critical angle for TIR can be changed using MLA. For the MLA etched OLEDs, most of the incident angles of light were smaller than the critical angle, because of curvature shape of the lenses, which led to most of the generated light being refracted from the outside surface. Con-sequently, light extraction efficiency i.e. EQE of the MLA engraved OLEDs was increased.

## 5 Conclusion

In summary, we designed efficient white OLEDs by engraving MLA on the top of the substrate layer. The basic OLED achieved an EQE of 14.45% for the ERI of 1.86. The addition of spheri-cal and elliptical (planoconvex and planoconcave) MLA on the substrate layer increased the EL intensity and EQE of the OLEDs significantly. All the designed OLEDs achieved higher EL intensity and EQE compared to basic OLED. Spherical planoconcave (Spherical-7) MLA incorporated OLEDs showed higher EL light intensity (36000 a.u.) and EQE (16.80%)

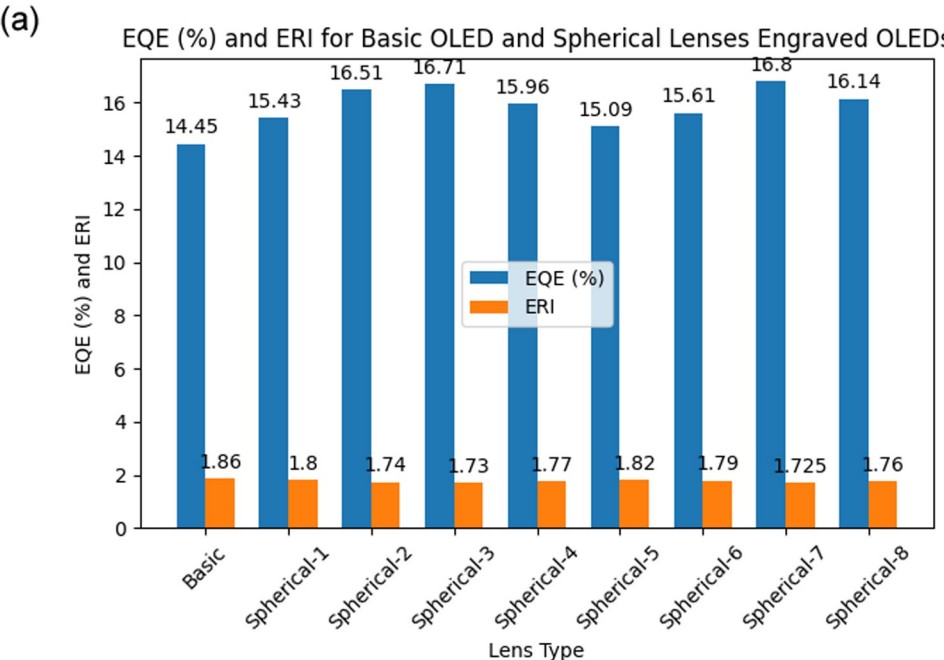

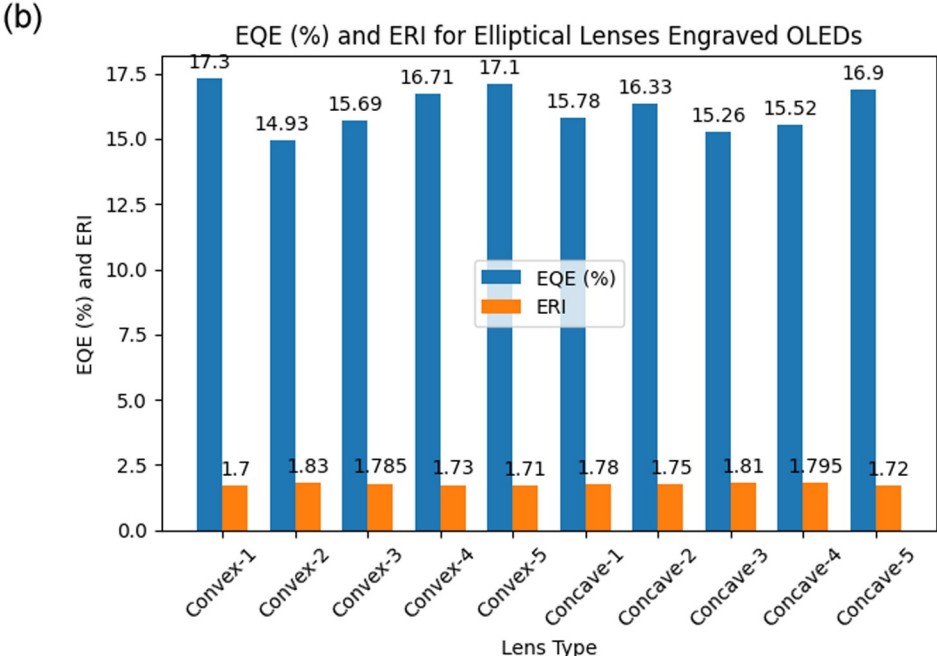

**Fig 18.** EQE and ERI for the OLEDs that have been designed: (a) Basic OLED and spherical MLA engraved OLEDs; (b): Elliptical MLA engraved OLEDs.

compared to spherical planoconvex MLA incorporated OLEDs. In addition, we optimized the value of radius/height of micro lenses as 0.6 $\mu$m to obtain smooth light distribution and higher EL light intensity of OLEDs. Among elliptical micro lenses, planoconvex MLA (Convex-1) engraved OLED showed the highest EQE compared to planoconcave MLA engraved OLEDs. Convex-1 MLA engraved OLED achieved the highest EQE of 17.30% for the ERI of 1.7 with

excellent light spreading and light distribution in the visible spectrum. In addition, Convex-1 MLA incorporated OLED demonstrated the highest EL light intensity of 38000 a.u. among the designed OLEDs which was 3.8 times higher compared to basic OLED model. We believe that our proposed light intensity and efficiency improvement technique can be considered as a potential contender for the advancement of efficient white OLEDs.

## Author Contributions

**Conceptualization:** Apurba Adhikary.

**Formal analysis:** Joy Bhuiya.

**Investigation:** Joy Bhuiya, Md. Bipul Hossain, MD Estihad Faysal.

**Methodology:** Apurba Adhikary, Md. Bipul Hossain, K. M. Aslam Uddin.

**Project administration:** K. M. Aslam Uddin.

**Supervision:** Abidur Rahaman, Anupam Kumar Bairagi.

**Validation:** K. M. Aslam Uddin, MD Estihad Faysal.

**Visualization:** Apurba Adhikary, Saydul Akbar Murad, MD Estihad Faysal.

**Writing – original draft:** Apurba Adhikary, Joy Bhuiya, Saydul Akbar Murad.

**Writing – review & editing:** Abidur Rahaman, Anupam Kumar Bairagi.

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
