## [Decision Letter · Decision Letter 0]

11 Apr 2022

PONE-D-22-07734Performance Evaluation of Micro Lens Arrays: Improvement of Light Intensity and Efficiency of White Organic Light Emitting DiodesPLOS ONE

Dear Dr. Bairagi,

Thank you for submitting your manuscript to PLOS ONE. After careful consideration, we feel that it has merit but does not fully meet PLOS ONE’s publication criteria as it currently stands. Therefore, we invite you to submit a revised version of the manuscript that addresses the points raised during the review process.

We look forward to receiving your revised manuscript.

Kind regards,

Yuan-Fong Chou Chau

Academic Editor

PLOS ONE

Journal Requirements:

Additional Editor Comments:

Dear Dr. Anupam Kumar Bairagi,

Thank you for submitting your manuscript to Adsorption Science & Technology. I have received comments from reviewers on your manuscript. Your paper should become acceptable for publication pending suitable major revision and modification of the article in light of the appended reviewer comments. When resubmitting your manuscript, please carefully consider all issues mentioned in the reviewers' comments, outline every change made a point by point, and provide suitable rebuttals for any comments not addressed.

Reviewers' comments:

Reviewer's Responses to Questions

**Comments to the Author**

1. Is the manuscript technically sound, and do the data support the conclusions?

Reviewer #1: Yes

Reviewer #2: Yes

Reviewer #3: Yes

2. Has the statistical analysis been performed appropriately and rigorously? 

Reviewer #1: Yes

Reviewer #2: Yes

Reviewer #3: Yes

3. Have the authors made all data underlying the findings in their manuscript fully available?

Reviewer #1: No

Reviewer #2: Yes

Reviewer #3: Yes

4. Is the manuscript presented in an intelligible fashion and written in standard English?

Reviewer #1: Yes

Reviewer #2: No

Reviewer #3: Yes

5. Review Comments to the Author

Reviewer #1: OLED light extraction technology using MLA is a rather outdated topic. Related theories are known.

In addition this paper only deals with simulation without providing actual device data.

"Acknowledgments" part is nonsensical.

This sumbission does not contribute to the field of OLEDs.

Reviewer #2: In this manuscript, the author proposes a unique method to improve light intensity and efficiency of white organic light emitting diodes (OLEDs) by engraving micro lens arrays (MLAs) on the outer face of the substrate layer. This work is meaningful. However, there are several points that need to be clarified before the paper can be accepted for publication.

1. This work mainly talks about the impact of MLA on the efficiency and light intensity of OLED devices, but rarely explains the reasons for the impact of MLA on OLEDs, especially in the introduction, although many examples are listed, the essential reasons for the impact of MLA on OLEDs should be highlighted. Why does MLA improve device efficiency? It would be better if we could clearly explain the essential impact of MLA. Please comment it.

2. At the end of line 31 in the introduction, it should be clear whether the captured light is 30% or 20%, or between 20% and 30%.

3. How stable is the OLED? Please comment it.

Reviewer #3: The authors proposed a unique method to improve light intensity and efficiency of white organic light emitting diodes (OLEDs) by engraving micro lens arrays (MLAs) on the outer face of the substrate layer. Technically, the authors employ the numerical method with apparent authority, and the results are interesting for the readers. However, the necessary physical mechanism and references are absent in the manuscript to explain the results. Therefore, I suggest accept this manuscript after major revision.

1.Please clarify in more detail the FDTD simulation setting, e.g., grid size, light source, perfectly matched layer (PML), polarization, and the width, height, and depth of layers used in Fig. 2.

2.EL intensity should be defined in the text..

3.On line 333-334, the mechanism of “all models of spherical planoconvex MLA engraved OLEDs significantly increased the EL light intensity compared to basic OLED” should be elucidated in more detail and cite the relevant references.

4.The advantage of the proposed structure should be discussed in more detail and compared to other type of approaches, e.g., “Localized surface plasmon resonance enhanced by the light-scattering property of silver nanoparticles for improved luminescence of polymer light-emitting diodes” (see Journal of Industrial and Engineering Chemistry, 2021, 103, pp. 283–291).

5.Please briefly address the fabrication issue of the proposed structure and cite the related articles.

6.The scale of color bar used in Figs. 5,6,7,9,10, 12,13,15, 16, should be the same and indicated the max. and min. values of color bars. One color bar used in one figure is enough.

6. PLOS authors have the option to publish the peer review history of their article (what does this mean?). If published, this will include your full peer review and any attached files.

Reviewer #1: No

Reviewer #2: No

Reviewer #3: No

---

## [Author Response · Author response to Decision Letter 0]

10 May 2022

All the comments of the reviewers have been resolved in the response letter and the corresponding changes are also made in the revised manuscript.

---

## [Decision Letter · Decision Letter 1]

16 May 2022

Performance Evaluation of Micro Lens Arrays: Improvement of Light Intensity and Efficiency of White Organic Light

Emitting Diodes

PONE-D-22-07734R1

Dear Dr. Bairagi,

We’re pleased to inform you that your manuscript has been judged scientifically suitable for publication and will be formally accepted for publication once it meets all outstanding technical requirements.

Kind regards,

Yuan-Fong Chou Chau

Academic Editor

PLOS ONE

Additional Editor Comments (optional):

I am pleased to accept this paper for publication in PLOS ONE.

Reviewers' comments:

Reviewer's Responses to Questions

**Comments to the Author**

1. If the authors have adequately addressed your comments raised in a previous round of review and you feel that this manuscript is now acceptable for publication, you may indicate that here to bypass the “Comments to the Author” section, enter your conflict of interest statement in the “Confidential to Editor” section, and submit your "Accept" recommendation.

Reviewer #2: All comments have been addressed

Reviewer #3: All comments have been addressed

2. Is the manuscript technically sound, and do the data support the conclusions?

Reviewer #2: Yes

Reviewer #3: Yes

3. Has the statistical analysis been performed appropriately and rigorously? 

Reviewer #2: (No Response)

Reviewer #3: Yes

4. Have the authors made all data underlying the findings in their manuscript fully available?

Reviewer #2: Yes

Reviewer #3: Yes

5. Is the manuscript presented in an intelligible fashion and written in standard English?

Reviewer #2: Yes

Reviewer #3: Yes

6. Review Comments to the Author

Reviewer #2: The authors have made substantial revisions according to reviewers' comments and all concerns have been addressed.

Reviewer #3: The authors have revised the manuscript according to my comments. This manuscript can now be accepted for publication.

7. PLOS authors have the option to publish the peer review history of their article (what does this mean?). If published, this will include your full peer review and any attached files.

Reviewer #2: No

Reviewer #3: No

---

## [Editor Report · Acceptance letter]

18 May 2022

PONE-D-22-07734R1 

Performance Evaluation of Micro Lens Arrays: Improvement of Light Intensity and Efficiency of White Organic Light Emitting Diodes 

Dear Dr. Bairagi:

I'm pleased to inform you that your manuscript has been deemed suitable for publication in PLOS ONE. Congratulations! Your manuscript is now with our production department. 

Kind regards, 

on behalf of

Dr. Yuan-Fong Chou Chau 

Academic Editor

PLOS ONE